# ADAPTIVE TEACHERS FOR AMORTIZED SAMPLERS

**Minsu Kim**[*]
Mila, KAIST

**Sanghyeok Choi**[*]
KAIST

**Taeyoung Yun**
KAIST

**Emmanuel Bengio**
Recursion

**Leo Feng**
Mila, Université de Montréal

**Jarrid Rector-Brooks**
Mila, Université de Montréal

**Sungsoo Ahn**
KAIST

**Jinkyoo Park**
KAIST

**Nikolay Malkin**
University of Edinburgh

**Yoshua Bengio**
Mila, Université de Montréal

## ABSTRACT

Amortized inference is the task of training a parametric model, such as a neural network, to approximate a distribution with a given unnormalized density where exact sampling is intractable. When sampling is implemented as a sequential decision-making process, reinforcement learning (RL) methods, such as generative flow networks, can be used to train the sampling policy. Off-policy RL training facilitates the discovery of diverse, high-reward candidates, but existing methods still face challenges in efficient exploration. We propose to use an adaptive training distribution (the Teacher) to guide the training of the primary amortized sampler (the Student). The Teacher, an auxiliary behavior model, is trained to sample high-loss regions of the Student and can generalize across unexplored modes, thereby enhancing mode coverage by providing an efficient training curriculum. We validate the effectiveness of this approach in a synthetic environment designed to present an exploration challenge, two diffusion-based sampling tasks, and four biochemical discovery tasks demonstrating its ability to improve sample efficiency and mode coverage. Source code is available at https://github.com/alstn12088/adaptive-teacher.

## 1 INTRODUCTION

Sampling from a complex distribution given its unnormalized density function is a fundamental problem in machine learning (Hinton, 2002; LeCun et al., 2006) and scientific discovery (Dellago et al., 1998; Noé et al., 2019). Amortized inference methods aim to fit a generative model that samples from a target distribution, possibly by a sequence of stochastic generation steps which is beneficial because it allows reusing a shared computational module for inference across multiple data points, as opposed to performing inference independently for each data point (Margossian & Blei, 2024). However, unlike for generative models trained from data, samples from the ground truth distribution may not be available. Multi-step sampling from an unnormalized density function with amortized inference can be achieved with reinforcement learning (RL) methods but raises the challenge of *exploration* – specifically, the ability to discover new modes of the target distribution during training. This is due to the intractable size of the sample space and the fact that only sampling from the generator itself would be oblivious to modes that the generator misses.

Just as with generative models trained on data, it is often more natural and beneficial to approximate the generation of objects as a sequence of decisions made by a policy rather than using a single parametric family, due to the multi-modal expressivity of hierarchical inference (*e.g.*, diffusion probabilistic models (Ho et al., 2020)). Sequential decision algorithms for amortized inference are unified by the theory of generative flow networks (GFlowNets; Bengio et al., 2021), which are a collection of off-policy RL methods (Tiapkin et al., 2024; Deleu et al., 2024). GFlowNets have been used for such amortized inference problems as natural-language and biological sequence design by token-by-token sequence generation (Jain et al., 2022; Shen et al., 2023; Hu et al., 2024), Bayesian inference over data structures (Deleu et al., 2022), molecular design by incremental addition of atoms or fragments (Bengio et al., 2021; Jain et al., 2023), or image refinement by a diffusion

---

[*]equal contribution, correspondence to {min-su, sanghyeok.choi}@kaist.ac.kr

Figure 1: Training an amortized sampler (Student) with an adaptive Teacher. **Left**: The behavior policy mixes Student, Teacher, and replay buffer policies to generate trajectories that train Student and store experiences. Teacher is updated based on Student's loss. **Right**: Student and Teacher distributions co-evolve, with Teacher targeting uncovered modes until Student converges to the target distribution.

process in continuous space (Venkatraman et al., 2024), *inter alia*. GFlowNets have shown success in the sequential sampling problems at scale due to their advantageous off-policy training ability (Malkin et al., 2023).

The prudent selection of training data is crucial to the success of such RL methods to model the full distribution faithfully, akin to the problem of active learning in supervised problems: to maximize sample efficiency, the most informative samples should be selected for training. To explore the full distribution, some applications of GFlowNets have either used exploration techniques borrowed from RL, such as noisy exploration (first used by Bengio et al., 2021), (prioritized) replay buffers (Deleu et al., 2022; Schaul, 2016; Vemgal et al., 2023), and delayed updates (Lau et al., 2023). Others have employed search techniques in the target space: MCMC and local search (Kim et al., 2024d;b; Sendera et al., 2024; Phillips & Cipcigan, 2024), genetic algorithms (Kim et al., 2024a), and exploiting samples from the target distribution (Zhang et al., 2022; Hu et al., 2023) when available.

However, challenges in mode coverage remain. All of the forementioned methods promote exploration through perturbation of the policy, *e.g.*, replaying the samples (Vemgal et al., 2023), augmenting the reward function (Pan et al., 2023b), and local search starting from generated samples (Kim et al., 2024d). These exploration methods focus on capturing the modes that are already near those generated by the current policy and can hardly capture the ones sufficiently separated from the already explored modes.

In this paper, we propose to explicitly explore the regions of high loss by introducing a **Teacher** model that guides the training of the primary, or **Student** sampler (Fig. 1). Here, we believe that trajectories with high loss are particularly informative for mode coverage, as they are likely to lead to regions of the target distribution that are either undersampled (dropped modes) or oversampled (collapsed modes). The **Teacher** is an adaptive behavior policy that is trained to sample target space regions where the **Student** model receives a high loss. In turn, the **Student** model is trained on samples from the **Teacher** model.

Our approach can be seen as amortizing an ideal prioritized experience replay (PER; Schaul, 2016) which samples high-loss objects from the entire sample space, instead of the finite-size replay buffer. Compared to (non-ideal) PER, the **Teacher** model has the potential to generalize across the high-loss regions of the **Student**, without regard for whether they have previously been sampled or discovered. In contrast, prioritized replay requires the poorly captured modes to have already been visited in order for the model to learn from them.

We test our method on a diverse set of domains where GFlowNets have been used, including discrete tasks (biological sequence design and molecular design) and continuous tasks (diffusion-based sampling benchmarks). Comprehensive experiments demonstrate that our algorithm is effective in improving mode coverage and training efficiency across all tasks.

## 2 PRELIMINARIES

We give a summary of GFlowNets as algorithms for amortized sampling by sequential decision making. For simplicity, this exposition is about discrete space, where GFlowNets were originally defined (Bengio et al., 2021), but GFlowNets have been extended to the case of continuous variables (Lahlou et al., 2023), which is conceptually similar. The reader is directed to Bengio et al. (2023) for an extended overview, Malkin et al. (2023) for an introduction focused on connections to hierarchical variational inference, and to Deleu et al. (2024) for a maximum-entropy RL point of view.

GFlowNets are policies in deterministic Markov decision processes (MDPs), trained so as to sample from a distribution over terminal states whose mass function is proportional to a given reward. The MDP is assumed to be represented as a finite directed acyclic graph $G = (\mathcal{S}, \mathcal{A})$, where $\mathcal{S}$ is the set of *states* and $(s \rightarrow s') \in \mathcal{A}$ if there is a possible action to be taken at state $s$ leading to state $s'$. A policy is then the same as a collection of distributions $P_F(\cdot \mid s)$ over the children[1] of every state $s$ that has at least one child. As in other deep RL methods, the policy could be a neural network $P_F(s' \mid s; \theta)$ taking a representation of the state $s$ as input and outputting the logits of a distribution over children $s'$. We will sometimes leave out the $\theta$ as implicit to lighten notation.

We assume the existence of a unique state $s_0 \in \mathcal{S}$, called the *initial state* that is not the child of any state. Conversely, states without children are called *terminal*, and the set of terminal states is denoted $\mathcal{X}$. A policy $P_F$ induces a distribution over *complete trajectories* – sequences $\tau = s_0 \rightarrow s_1 \rightarrow \cdots \rightarrow s_n$ with $s_n \in \mathcal{X}$ – which can be sampled by starting at $s_0$ and iteratively transitioning to child states sampled according to $P_F$ until a terminal state is reached. This in turn a *terminating distribution* $P_F^\top$ over $\mathcal{X}$, which is the marginal distribution over the final states of trajectories sampled in this way. To be precise,

$$P_F^\top(x) = \sum_{\tau \leadsto x} P_F(\tau), \quad P_F(\tau = (s_0 \rightarrow \cdots \rightarrow s_n)) := \prod_{i=0}^{n-1} P_F(s_{i+1} \mid s_i), \tag{1}$$

where $x \in \mathcal{X}$ and $\tau \leadsto x$ indicates that the sum is restricted to trajectories $\tau$ whose last state is $x$.

Let $R : \mathcal{X} \rightarrow \mathbb{R}_{>0}$ be a function on the terminal states, called the *reward function*, and set $Z := \sum_{x \in \mathcal{X}} R(x)$ as the normalization constant. We would like to train $P_F$ so as to make $P_F^\top(x) = R(x)/Z$ for all $x \in \mathcal{X}$, *i.e.*, to make $P_F$ sample terminal states with probability proportional to the reward.

Because the sum in (1) may be intractably large (if many trajectories could lead to the same $x$), achieving this requires introducing auxiliary objects into the optimization. One popular option is to use the trajectory balance (TB) objective (Malkin et al., 2022). To train a model with TB, one introduces an additional *backward policy* $P_B(\cdot \mid \cdot)$, which is a collection of distributions over the parents of every noninitial state (*i.e.*, a policy on the reverse MDP, which can be either fixed – as in our case – or learned), as well as an estimate of the total reward $Z_\theta$ (usually parametrized in the log domain, making $\log Z_\theta$ a learnable parameter). We define the TB discrepancy for a trajectory $\tau$ with final state $x$ by

$$\delta(\tau; \theta) := \underbrace{[\log R(x) + \log P_B(\tau \mid x)]}_{\text{backward flow}} - \underbrace{[\log Z_\theta + \log P_F(\tau; \theta)]}_{\text{forward flow}}, \tag{2}$$

where $P_B(\tau \mid x; \theta)$ is defined analogously to (1), by

$$P_B(\tau = (s_0 \rightarrow \cdots \rightarrow s_n) \mid x) = \prod_{i=0}^{n-1} P_B(s_i \mid s_{i+1}). \tag{3}$$

It can be shown that if $\delta(\tau; \theta) = 0$ for all trajectories $\tau$, then $Z_\theta = Z$ and $P_F^\top(x) = R(x)/Z$ for all $x$, meaning that $P_F$ solves the sampling problem. Intuitively, this is the case because the reward distribution and $P_B$ would then determine the same distribution over trajectories as $P_F$, but factorized in reverse order. One thus attempts to enforce this by minimizing a loss, such as $\delta(\tau; \theta)^2$, on trajectories $\tau$ sampled from some behaviour policy $\pi$. (If $\pi$ is the current policy $P_F$ itself, the optimization is said to be *on-policy*, otherwise *off-policy*.)

We note that there exist other training procedures that use learned estimators and loss functions associated with individual states or transitions, rather than full trajectories, such as detailed balance (DB; Bengio et al., 2023) and subtrajectory balance (SubTB; Madan et al., 2023), which all have advantages under certain conditions. This paper mostly focuses on TB due to its simplicity and popularity as a default choice in the literature; we investigate DB in Appendix F to show our method's flexibility over objective functions.

---

[1] If $(s \rightarrow s') \in \mathcal{A}$, then $s'$ is a *child* of $s$; the converse relation is called *parent*.

## 3 THE TEACHER: AN ADAPTIVE TRAINING DISTRIBUTION

In this section, we introduce the **Teacher**, which is a secondary GFlowNet designed to enhance the efficiency of off-policy training for the primary, or **Student**, GFlowNet. The two GFlowNets share the same state and action space, but have different rewards.

The Teacher's role is to generate an adaptive training distribution for the Student, aiming to sample trajectories that yield high loss for the Student. Intuitively, samples with high loss tend to be less (or never) visited by the Student, implying high probability of the samples being in the unexplored modes. To this end, we train the Teacher with GFlowNet objective for amortization of sampling trajectories with high loss. Note that the Student's target distribution does not depend on the Teacher, but the Teacher's target distribution depends on the Student.

We henceforth denote the parameters of the Student GFlowNet by $\theta$ and those of the Teacher by $\phi$.

### 3.1 REWARD DESIGN FOR TEACHER

We define the reward function for the Teacher using the TB loss of the Student, $\delta(\tau; \theta)^2$. In basic form, we could define the Teacher's reward as

$$\log R_{\text{Teacher}}^{\text{basic}}(x; \theta) = \mathbb{E}_{P_B(\tau|x;\theta)} \left[ \log \left( \delta \left( \tau; \theta \right)^2 \right) \right]. \tag{4}$$

In Eq. (4), the Student's loss is marginalized over trajectories $\tau$ in the log domain over the backward policy $P_B(\tau \mid x; \theta)$ of the Student, given a terminal state $x$. This is because we aim to train the Teacher as a sampler of terminal states, although what we obtain when training the Student is a *trajectory-level* error $\delta(\tau; \theta)$, which we need to convert into a function of the terminal state $x$ only to form the Teacher's reward. Having the Teacher model terminal states $x$ rather than full trajectories $\tau$ is motivated by the desire to obtain full mode coverage in the space of terminal states, but not necessarily in the space of trajectories that lead to these terminal states. In practice, this expectation is estimated using Monte Carlo sampling with a single sample $\tau \sim P_B(\tau \mid x; \theta)$, relying on the fact that stochastic gradient descent training of the Teacher will automatically average out the variability resulting from this sampling. In fact, this gives an unbiased estimator of the gradient if training with the full expectation (see, *e.g.*, Deleu et al., 2022; Bengio et al., 2023).

We propose two modifications to (4) to facilitate mode discovery.

**Favoring undersampled regions.** First, we hypothesize that because the Teacher should encourage the Student to discover unvisited modes, it should favor regions of the state space where the target density exceeds the Student's sampling probability. To this end, we increase the weight of the Teacher's reward for states where the backward flow exceeds the forward flow (cf. (2)), while adding a smoothing constant $\epsilon$:

$$\log R_{\text{Teacher}}^{\text{weighted}}(x; \theta) = \mathbb{E}_{P_B(\tau|x;\theta)} \left[ \log \left( \epsilon + \left( 1 + C \mathbb{I}_{\delta(\tau;\theta)>0} \right) \delta \left( \tau; \theta \right)^2 \right) \right], \tag{5}$$

The weighted term $(1 + C \mathbb{I}_{\delta(\tau;\theta)>0})$ gives additional weight when the TB discrepancy is positive, which indicates the Student is undersampling a high-rewarded terminal sample. Here, $C > 0$ represents the weighting constant, which we set to $C = 19$ for every task; see Appendix E.3 for ablation study on our choice of $C$.

**Reward mixing.** To ensure the Teacher covers the missing modes (the high-reward regions that the Student missed), it is important to focus the Teacher's search space more on the high-reward regions than the low-reward ones. This approach helps target both high-loss and high-reward areas effectively. To achieve this, we propose to mix the reward (5), which is based on the Student's loss, with the Student's log reward:

$$\log R_{\text{Teacher}}(x) := \log R_{\text{Teacher}}^{\text{weighted}}(x) + \alpha \log R(x). \tag{6}$$

This approach encourages the Teacher to sample regions with both high loss and high reward. Here the mixing constant $\alpha$ is a hyperparameter that trades off between high loss and high reward; see Appendix E.4 for analysis of the effect of the choice of $\alpha$.

---

**Algorithm 1** Teacher-Student Training of GFlowNets

---

1: $Q_{\text{buffer}} \leftarrow \emptyset$      ▷ *Initialize replay buffer with queue structure*

2: **for** $t = 1, \ldots, T$ **do**      ▷ *Iteration of training rounds*

3:      Select behavior policy $P_{\beta}(\tau) = \begin{cases} P_F(\tau; \theta), & \text{if select Student} \\ P_F(\tau; \phi), & \text{if select Teacher} \\ P_B(\tau|x)P(x|Q_{\text{buffer}}), & \text{if select Prioritized Buffer.} \end{cases}$

4:      Sample trajectories $\tau_1, \ldots, \tau_B \sim P_{\beta}(\tau)$.      ▷ *Exploration*

5:      (Optional) Refine trajectories using local search.

6:      Compute rewards: $R(x_1), \ldots, R(x_B)$.

7:      Compute TB discrepancy of Student: $\delta(\tau_1; \theta), \ldots, \delta(\tau_B; \theta)$.

8:      Compute $R_{\text{Teacher}}(x_1), \ldots, R_{\text{Teacher}}(x_B)$ using $\{R(x_i)\}_{i=1}^B$ and $\{\delta(\tau_i; \theta)\}_{i=1}^B$.

9:      Compute TB discrepancy of Teacher: $\delta(\tau_1; \phi), \ldots, \delta(\tau_B; \phi)$.

10:      Update Student parameters: $\theta \leftarrow \text{Optimizer} \left( \frac{1}{B} \sum_{i=1}^B \delta(\tau_i; \theta)^2 \right)$      ▷ *Student training*

11:      Update Teacher parameters: $\phi \leftarrow \text{Optimizer} \left( \frac{1}{B} \sum_{i=1}^B \delta(\tau_i; \phi)^2 \right)$      ▷ *Teacher training*

12:      Add experiences to buffer: $Q_{\text{buffer}} \leftarrow Q_{\text{buffer}} \cup \{x_i, R(x_i), R_{\text{Teacher}}(x_i)\}_{i=1}^B$

13: **end for**

---

## 3.2 Jointly training Teacher and Student

Using the $R_{\text{Teacher}}(x; \theta)$, the training process is a joint optimization of the Teacher parameters $\phi$ and the Student parameters $\theta$ to jointly minimize the following loss functions:

$$\mathcal{L}_{\text{Student}}(\tau; \theta) = \delta(\tau; \theta)^2 = \left( \log \frac{Z_\theta P_F(\tau; \theta)}{R(x) P_B(\tau \mid x)} \right)^2, \tag{7}$$

$$\mathcal{L}_{\text{Teacher}}(\tau; \phi) = \delta_{\text{Teacher}}(\tau; \phi)^2 = \left( \log \frac{Z_\phi P_F(\tau; \phi)}{R_{\text{Teacher}}(x; \theta) P_B(\tau \mid x)} \right)^2, \tag{8}$$

Notice that (7) is the loss for regular TB training of the Student, while (8) relies on the Student's loss to provide a reward for the Teacher via (6).

To simplify the training process, we adopt in our experiments a fixed backward policy $P_B(\tau \mid x)$, without trainable parameters, which is used by both the Teacher and the Student.

**Behavior policy for joint optimization.** Algorithm 1 describes the Teacher-Student training procedure, which simultaneously minimizes the loss functions of both the Teacher and the Student. In line 3, we select the behavior policy by choosing either the Student, the Teacher, or a prioritized buffer, ensuring that all three are sufficiently utilized (see Appendix C for details on how they are chosen). Given a terminal state $x$ sampled from $P(x \mid Q_{\text{buffer}})$, we then generate a trajectory $\tau$ using the backward policy $P_B(\tau \mid x)$. This approach is similar to previous works (Shen et al., 2023; Sendera et al., 2024), which store only the terminal states $x$ in the buffer and sample trajectories $\tau$ using $P_B$.

The behavior policy can produce an adaptive distribution of trajectories $\tau$ with respect to the Student's learning state $\theta$ because the Teacher iteratively focuses on high-loss trajectories of the Student during training. This adaptivity is hypothesised to result in highly effective training for the Student.

**Existence of a stationary point of training process.** The joint optimization for the parameters $\phi$ and $\theta$ over the support of $P_{\beta}(\tau)$ has a stationary point where the Student GFlowNet becomes an exact sampler and the Teacher GFlowNet samples proportional to $\epsilon R(x)^\alpha$; see Prop. 1 in Appendix B.

## 3.3 Mitigating non-stationarity with local search

Joint optimization of the parameters $\phi, \theta$ with a non-stationary target $R_{\text{Teacher}}(x; \theta)$ poses significant challenges as the Teacher's reward is nonstationary, evolving as the Student learns. To address this issue, we use a local search method (Line 5) that locally optimizes $R_{\text{Teacher}}(x; \theta)$ based on the Teacher's samples. We expect the dynamic nature of $R_{\text{Teacher}}(x; \theta)$ to be effectively managed by such search, as the Teacher's main role is to generalize to modes poorly modeled by the Student, while the search helps the Teacher track the *local* changes in the Student's loss landscape.

Local search using a kernel defined by the policies – first used by Zhang et al. (2022); Hu et al. (2023) and extensively studied by Kim et al. (2024d) – involves iteratively backtracking trajectories and reconstructing them to produce new samples. The method consists of the following steps:

1. **Backtracking**: Starting from a terminal state $x$, we backtrack to an intermediate state $s$ using the backward policy $P_B$, denoted as $(x \dashrightarrow \ldots \dashrightarrow s)$.
2. **Reconstruction**: From the intermediate state $s$, we reconstruct a new terminal state $x'$ using the Teacher's forward policy $P_F$, represented as $(s \rightarrow \ldots \rightarrow x')$.
3. **Accept or reject**: We accept the new sample $x'$ in place of $x$ with acceptance probability $A$.

The acceptance probability $A$ can be determined using either a stochastic Metropolis-Hastings (MH) approach or a deterministic ascent criterion (see Kim et al. (2024d) for details). This process is repeated iteratively to progressively improve the samples so that their reward better matches (with MH) or locally maximizes (with the deterministic ascent version) the target reward function. Ultimately, we use the enhanced sample $x'$ to train both the Teacher and the Student by generating trajectories $\tau \sim P_B(\tau \mid x')$ and taking gradient steps on the losses (7) and (8).

## 4 RELATED WORK

**GFlowNets.** GFlowNets were originally introduced by Bengio et al. (2021) and extensively extended by Bengio et al. (2023). They aim to develop a sequential decision-making policy with a form of deep reinforcement learning that aims at sampling from the unnormalized density associated with a positive reward function. Aiming to improve credit assignment over long trajectories, Malkin et al. (2022) introduced the trajectory balance (TB) objective mainly used in this paper. Building on this, Madan et al. (2023) introduced a mixing scheme that combines losses associated with subtrajectories, trading off the lower variance of DB with the lower bias of TB, Pan et al. (2023a) studied inductive biases that use partial reward information, and Jang et al. (2024b) extended this idea to learnable reward shaping schemes. Shen et al. (2023) and Jang et al. (2024a) studied auxiliary losses for better training of the backward policy.

Orthogonal to those studies, other works focus on improving off-policy training. Deleu et al. (2022); Shen et al. (2023); Vemgal et al. (2023) studied the use of replay buffers in GFlowNets to enhance sample efficiency. Kim et al. (2024d;a;b); Sendera et al. (2024) investigated local search methods to guide GFlowNets toward high-reward regions. Kim et al. (2024c) proposed to adjust the exploration-exploitation trade-off via amortized conditioning on reward temperature. Similarly, Lau et al. (2024) introduced a method that mixes Deep Q-Networks (Mnih, 2013) (exploitation) with GFlowNets (exploration) to balance the exploration-exploitation trade-off. Our proposed method is also an off-policy training approach for GFlowNets. In contrast to the methods above, which use off-policy training to focus GFlowNets on high-reward regions, our method aims to address missing modes and underexplored regions. Note that the aforementioned reward-seeking off-policy methods are complementary to our approach; for example, local search with a Teacher is studied in §5.1.

While the above algorithmic work mostly concerns discrete space, Lahlou et al. (2023) introduced the theory of GFlowNets in continuous space, leading to subsequent work on diffusion samplers (Zhang et al., 2024; Sendera et al., 2024), posterior sampling under diffusion priors (Venkatraman et al., 2024), and applications to molecular dynamics (Seong et al., 2024). Our proposed algorithms are effective in both discrete and continuous space (§5.2).

**Adaptive training distributions.** Adaptive training distributions are essential techniques in deep learning, ensuring that the training data evolves appropriately during model training. For instance, curriculum learning methods (Bengio et al., 2009) schedule the difficulty of training tasks by gradually increasing from easy to hard, thereby facilitating more efficient model training. These methods are also widely applied in reinforcement learning, *e.g.*, Narvekar et al. (2020).

Active learning (Gal et al., 2017), which is usually built for supervised learning, also falls into this category, where the training dataset actively changes to discover better strategies. Especially, uncertainty sampling-based methods (Sener & Savarese, 2018; Yoo & Kweon, 2019; Kirsch et al., 2019; Ash et al., 2020) that prioritize sampling data points having high predictive uncertainty of the classifier are relevant to our idea. The key difference is that our method is built for reinforcement learning, where we do not rely on the predictive uncertainty of a classifier but on the loss value defined by the compositional policy.

Our work is also relevant to few-shot experimental design (Wang et al., 2024), as our Teacher plays the similar role as the entropy-regularized adversary that generates tasks.

Table 1: Evaluation results on deceptive grid worlds with dimension $d$ and grid length $H$. The number of modes discovered (# modes) and empirical $L_1$ distance between target and sampled distributions are reported as mean ±standard deviation over five runs. The $L_1$ distances are scaled appropriately for readability.

| Grid config. → | $d = 2, H = 128$ | | $d = 2, H = 256$ | | $d = 4, H = 16$ | | $d = 4, H = 32$ | |
|---|---|---|---|---|---|---|---|---|
| Algorithm ↓ Metric → | # modes (↑) | $L_1 \times 10^{-5}$ (↓) | # modes (↑) | $L_1 \times 10^{-5}$ (↓) | # modes (↑) | $L_1 \times 10^{-5}$ (↓) | # modes (↑) | $L_1 \times 10^{-6}$ (↓) |
| TB (on-policy →) | 645.4 ± 41.5 | 2.20 ± 0.58 | 733.6 ± 25.1 | 1.74 ± 0.04 | 6.6 ± 2.5 | 1.027 ± 0.012 | 16.6 ± 4.8 | 1.635 ± 0.000 |
| + $\epsilon$-expl. | 555.2 ± 66.2 | 3.59 ± 0.66 | 672.6 ± 16.3 | 1.75 ± 0.02 | 6.6 ± 4.2 | 1.030 ± 0.006 | 24.2 ± 3.2 | **1.634** ± 0.000 |
| + GAFN | 675.4 ± 0.5 | **1.66** ± 0.11 | 1044.8 ± 276.4 | 1.55 ± 0.17 | 11.8 ± 3.9 | 1.057 ± 0.022 | 24.6 ± 7.4 | 1.664 ± 0.002 |
| + PRT | **676.0** ± 0.0 | 4.54 ± 0.14 | 2165.2 ± 64.5 | 1.55 ± 0.05 | 38.8 ± 10.0 | 1.097 ± 0.006 | 120.4 ± 19.1 | 1.648 ± 0.001 |
| + PER | 669.0 ± 3.8 | 4.88 ± 0.44 | 2055.2 ± 56.3 | 1.71 ± 0.08 | 16.0 ± 2.8 | 1.129 ± 0.034 | 46.6 ± 14.6 | 1.639 ± 0.001 |
| + Teacher (*ours*) | **676.0** ± 0.0 | 2.13 ± 0.18 | **2452.6** ± 21.7 | **0.94** ± 0.03 | **51.4** ± 4.0 | **1.019** ± 0.016 | **246.6** ± 14.7 | **1.634** ± 0.001 |

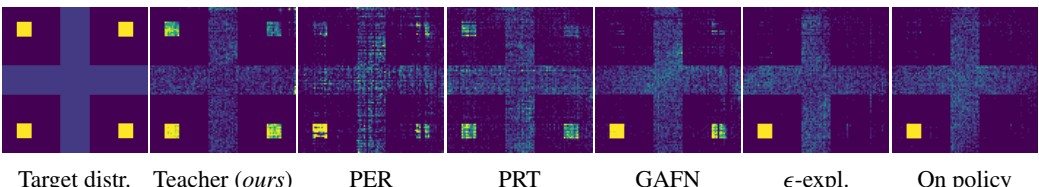

Target distr. Teacher (*ours*) PER PRT GAFN $\epsilon$-expl. On policy

Figure 2: Empirical distribution plots of $10^5$ test samples from policies on the ($d = 2, H = 256$) grid.

## 5 EXPERIMENTS

This section provides empirical validation of our method. Our primary goal is to demonstrate that our approach is beneficial over other off-policy training methods for trajectory balance (TB), a representative learning objective for amortized samplers. We also aim to show that our method can effectively be integrated with existing off-policy search methods, such as local search and replay buffer techniques. We benchmark our approach on three major tasks: two discrete tasks (§5.1 and 5.3) and one continuous task (§5.2). We also include three ablation studies and a pilot experiment on integrating local search in Appendix E. Additionally, we tested the versatility of our method by adopting another training objective, detailed balance (DB; Bengio et al., 2023) in Appendix F.

### 5.1 DECEPTIVE GRID WORLD

**Setting.** The deceptive grid world is a synthetic environment modified from the grid task introduced by Bengio et al. (2021). It consists of a $d$-dimensional hypercube of side length $H$, resulting in a search space of size $O(H^d)$. The agent starts from the origin (position $\mathbf{0}$) and can only move in directions that increase a coordinate by 1 or terminate to receive the reward. The reward of each terminal state $x = (x_1, \ldots, x_d)$ is given by

$$R(x) = R_0 + R_1 \prod_{i=1}^{d} \mathbb{I}\left[\left|\frac{x_i}{H-1} - 0.5\right| < 0.1\right] + R_2 \prod_{i=1}^{d} \mathbb{I}\left[\left|\frac{x_i}{H-1} - 0.5\right| \in (0.3, 0.4)\right]. \quad (9)$$

The modes with the highest rewards ($R_2 + R_0$) are surrounded by walls with low rewards ($R_0$). In between them are deceptive regions offering relatively high rewards ($R_1 + R_0$), which can lure the agent into getting trapped. We set $R_0 = 10^{-5}$, $R_1 = 0.1$, and $R_2 = 2$. Following previous works, we use the number of modes discovered and the empirical $L_1$ distance from the target distribution as evaluation metrics. See Appendix D.1 for more details about the settings.

**Baselines.** We compare our method with on-policy TB and TB with off-policy exploration methods, such as $\epsilon$-exploration (Bengio et al., 2021), and GAFN (Pan et al., 2023b), along with a baseline using a replay buffer prioritizing rewards (PRT) or Teacher rewards inspired by Prioritized Experience Replay (PER; Schaul, 2016). PER can be seen as a non-amortized version of our method.

**Results.** Table 1 summarizes the results. TB with Teacher consistently outperforms baselines. The significant margin in the number of modes discovered in the larger-scale setting indicates that the Teacher effectively guides the Student to visit undiscovered modes. Please refer to Appendix E.1 for comprehensive results.

**Effects of local search.** We assess the local search (LS) effect (§3.3) on a ($d = 4, H = 32$) grid. The Teacher's local search uses the Teacher's reward $R_{\text{Teacher}}$ for acceptance, compared to two baselines – on-policy TB and TB with PER – which use the Student's reward $R$. Furthermore, local search accelerates mode discovery, likely by reducing the sensitivity to nonstationarity in Teacher learning (See Appendix E.5).

Table 2: Evaluation on multimodal continuous sampling tasks. Log-partition function estimation errors (evidence lower bound (ELBO), importance sampled ELBO (ELBO-IS), evidence upper bound (EUBO)) and 2-Wasserstein distances ($W_2^2$) to target samples are reported as mean$\pm$std over five runs. We compare MCMC methods (SMC, GGNS), a differentiable simulation method (PIS), and GFlowNets trained using the TB objective with various off-policy strategies, such as loss prioritized replay (PER) and reward prioritized replay (PRT).

| Energy → | 25GMM ($d = 2$, $\log Z = 0$, $W_2^2 = 0.29$) | | | | Manywell ($d = 32$, $\log Z = 164.6956753$, $W_2^2 = 5.36$) | | | |
|---|---|---|---|---|---|---|---|---|
| Algorithm ↓ Metric → | ELBO (↑) | ELBO-IS (↑) | EUBO (↓) | $W_2^2$ (↓) | ELBO (↑) | ELBO-IS (↑) | EUBO (↓) | $W_2^2$ (↓) |
| SMC | | - 0.569$_{\pm0.010}$ | | 0.86$_{\pm0.10}$ | | 149.706$_{\pm1.078}$ | | 8.28$_{\pm0.32}$ |
| GGNS | | - 0.016$_{\pm0.042}$ | | 1.19$_{\pm0.17}$ | | 164.404$_{\pm0.454}$ | | 6.51$_{\pm0.32}$ |
| PIS | -1.192$_{\pm0.177}$ | -1.192$_{\pm0.176}$ | 26.733$_{\pm5.107}$ | 4.95$_{\pm0.73}$ | 160.516$_{\pm1.025}$ | 162.017$_{\pm0.980}$ | 581.464$_{\pm240.916}$ | 6.15$_{\pm0.01}$ |
| TB (on-policy →) | -1.105$_{\pm0.007}$ | -1.008$_{\pm0.009}$ | 18.321$_{\pm0.932}$ | 4.64$_{\pm0.01}$ | 161.048$_{\pm0.036}$ | 162.019$_{\pm0.645}$ | 427.850$_{\pm80.082}$ | 6.15$_{\pm0.01}$ |
| + $\epsilon$-expl. | -1.056$_{\pm0.117}$ | -0.956$_{\pm0.118}$ | 15.135$_{\pm1.861}$ | 4.58$_{\pm0.08}$ | 161.064$_{\pm0.036}$ | 162.008$_{\pm0.062}$ | 355.787$_{\pm4.761}$ | 6.15$_{\pm4.761}$ |
| + PRT | -0.750$_{\pm0.138}$ | -0.640$_{\pm0.144}$ | 12.103$_{\pm2.273}$ | 4.28$_{\pm0.59}$ | 161.071$_{\pm0.085}$ | 161.998$_{\pm0.111}$ | 379.623$_{\pm77.409}$ | 6.14$_{\pm0.03}$ |
| + PER | -0.282$_{\pm0.158}$ | -0.147$_{\pm0.162}$ | 1.833$_{\pm2.366}$ | 1.87$_{\pm1.23}$ | 161.537$_{\pm0.186}$ | 162.582$_{\pm0.268}$ | 210.440$_{\pm6.888}$ | 5.91$_{\pm0.08}$ |
| + Teacher (*ours*) | **-0.137**$_{\pm0.004}$ | **-0.005**$_{\pm0.007}$ | **0.115**$_{\pm0.009}$ | **0.86**$_{\pm0.07}$ | **163.484**$_{\pm0.049}$ | **164.676**$_{\pm0.048}$ | **165.800**$_{\pm0.045}$ | **5.46**$_{\pm0.01}$ |

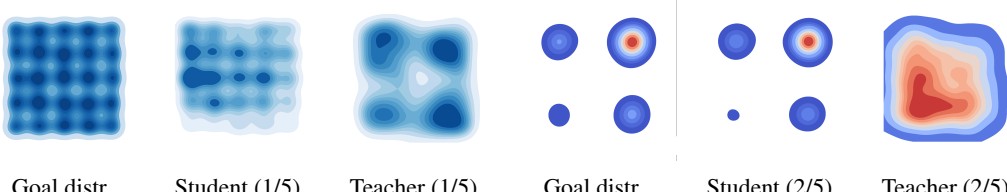

| Target distr. | Teacher (*ours*) | PER | PRT | $\epsilon$-expl. | On policy |
|---|---|---|---|---|---|

Figure 3: Samples from trained models on the Manywell task (projected onto the first two dimensions).

| Goal distr. | Student (1/5) | Teacher (1/5) | Goal distr. | Student (2/5) | Teacher (2/5) |
|---|---|---|---|---|---|

Figure 4: KDE plots for 25GMM (left three) and Manywell (right three) at intermediate states of training. The Student (*ratio*) indicates the fraction of total training steps completed. The Teacher adaptively adjusts the training distribution in response to the modes that the Student is missing.

## 5.2 DIFFUSION SAMPLING

**Setting.** The task of learning a *diffusion sampler* is to invert a diffusion process in order to sample the target density function. These experiments largely follow the setup of Sendera et al. (2024). Here, we aim to model a distribution over trajectories $\tau = (0 = x_0 \rightarrow x_{\Delta t} \rightarrow x_{2\Delta t} \rightarrow \ldots \rightarrow x_1)$ (with $\Delta t = \frac{1}{T}$; for us, $T = 100$), so that $x_1$ is distributed according to an unnormalized density function $p(x_1) \propto R(x_1) = e^{-\mathcal{E}(x_1)}$. The transitions are parametrized as to the Euler-Maruyama discretization of a neural stochastic differential equation (Tzen & Raginsky, 2019): the sampler begins at the initial state $(0, t = 0)$, and the transition from $x_t$ to $x_{t+\Delta t}$ is sampled from an appropriately scaled Gaussian with mean given by a trained model taking $x_t$ and $t$ as input. The detailed setting and policy parametrization are described in Appendix D.2.

The main challenge in this task to capture the multimodality of $R(x_1)$ without having access to samples from target distribution during training, which is difficult when there are many well-separated modes. It is not possible to apply the forward KL (*i.e.*, log-likelihood variational bound) objectives typically used for diffusion models (Song et al., 2021), since target distribution datapoints are not available; thus *exploration* is crucial to mode discovery.

In this work, we benchmark diffusion samplers on two established tasks in the diffusion sampling literature: a 2-dimensional Gaussian mixture with 25 modes (25GMM) and a 32-dimensional Manywell distribution. When performing off-policy exploration, we assume a black-box property for the energy function $\mathcal{E}$, where $\nabla\mathcal{E}$ is not accessible, similar to settings in reinforcement learning. This assumption is meant to mimic a common setting in scientific discovery, where we may have black-box energies requiring expensive simulations to compute; we note that for these tasks, effective methods that use the energy gradient exist (see Sendera et al. (2024)).

**Baselines.** We consider two representative MCMC baselines: Sequential Monte Carlo (SMC) and the state-of-the-art GGNS (Lemos et al., 2024). We also include the simulation-based continuous

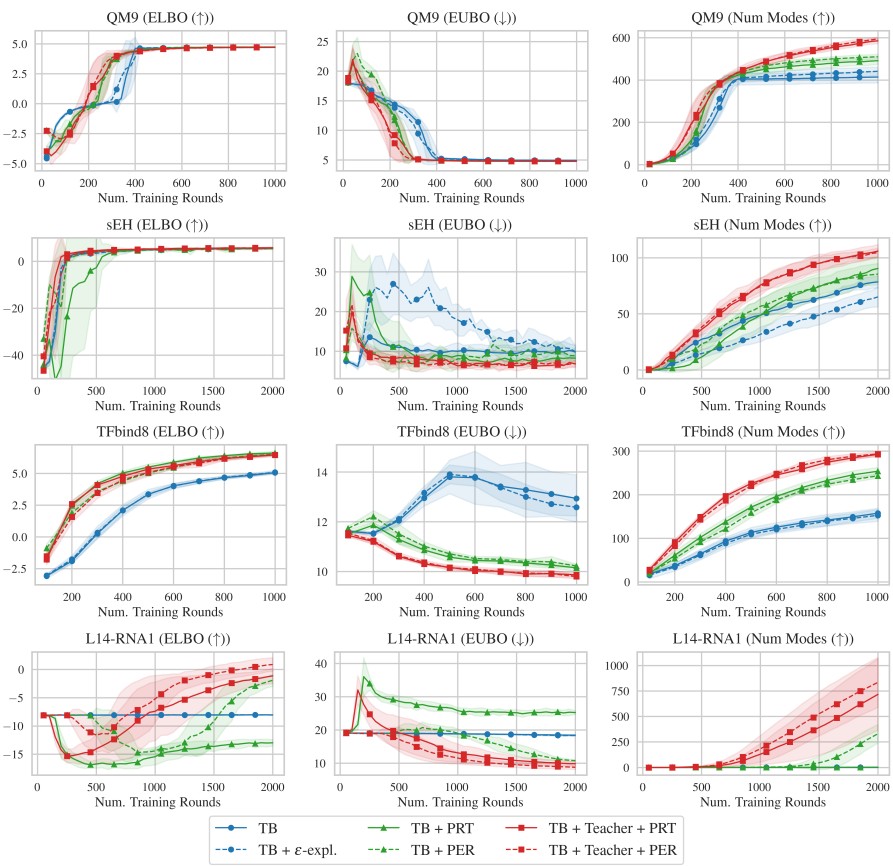

Figure 5: Training graphs for molecule design (QM9, sEH) and biological sequence design (TFbind8, L14-RNA1) tasks. Mean and standard deviation over five runs are shown.

stochastic control method Path Integral Sampler (PIS; Zhang & Chen, 2022). The major baselines are off-policy training methods based on trajectory balance (TB). We include on-policy TB in continuous space and the $\epsilon$-exploration method introduced by Malkin et al. (2023); Lahlou et al. (2023), which adds additional Gaussian noise at least policy sampling step during training, as main baselines. Additionally, we compare with two replay buffer methods combined with TB: one that prioritizes reward (PRT) as studied by Sendera et al. (2024), and another that prioritizes loss (PER) (Schaul, 2016). The gradient-based local search introduced by Sendera et al. (2024), while effective, is excluded as it requires access to $\nabla \mathcal{E}(x)$ for the search; but we also study potential integration of the Teacher with these techniques in Appendix E.5.

**Results.**    As shown in Table 2, on-policy TB and PIS yield similar performance, consistent with the fact that they have identical expected gradients (Malkin et al., 2023; Lahlou et al., 2023). This suggests that TB could benefit from additional off-policy methods for improvement. Indeed, the $\epsilon$-exploration techniques improve slightly over on-policy methods. PRT provides larger benefits on 25GMM but shows no meaningful benefits on Manywell. PER offers significant improvements over on-policy methods compared to others. Our method, Teacher, achieves the highest results across all metrics, including the Evidence Lower Bound (ELBO), Importance-Sampled ELBO (ELBO-IS), Evidence Upper Bound (EUBO) (see Appendix D for detailed definitions of those metrics). Especially in EUBO, a metric suggested by Blessing et al. (2024) to measure mode coverage, baseline methods face significant challenges. This indicates that existing methods struggle to perform proper *exploration* across modes, as confirmed by the sample plots in Fig. 3. Our method offers clear advantages on EUBO metrics.

In Fig. 4, we depict the training dynamics of the Teacher and Student by plotting kernel density estimates of their samples midway through training. This figure illustrates the mechanism by which the Teacher promotes mode discovery by the Student. As the Student struggles to find some modes, especially those with lower rewards than others, the Teacher puts high probability on them, encouraging the Student to reduce its loss in those regions of the space.

## 5.3 BIOLOGICAL AND CHEMICAL DISCOVERY

**Setting.** GFlowNets have been used to generate biological and chemical structures by sequentially adding predefined substructures. In molecules, the actions add atoms or fragments; in biological sequences, nucleotides or amino acids. We aim to match a target probability distribution over structures given by some proxy reward model. Following Shen et al. (2023), we benchmark the number of discovered modes, as well as probabilistic metrics like ELBO and EUBO. We study four biological and chemical discovery problems, following Shen et al. (2023); Kim et al. (2024d):

- **QM9.** The objects being sampled are small molecular graphs. Molecules are generated using 12 building blocks with 2 stems, and each molecule contains 5 blocks. The reward function is a HOMO-LUMO gap on the target transcription factor, which is obtained via a pre-trained MXMNet proxy from Zhang et al. (2020). We use a reward exponent of 5. We define modes as the top 0.5% quantile of $R(x)$.
- **sEH.** The generated objects are molecular graphs. Molecules are built using 18 blocks with 2 stems and 6 blocks per molecule. The reward function is a binding affinity to soluble epoxide hydrolase (sEH), which is provided by the pre-trained proxy model from Bengio et al. (2021). We use a reward exponent of 6. We define modes as the top 0.01% quantile of $R(x)$, with filtering to exclude candidates too similar based on Tanimoto similarity, following Kim et al. (2024c).
- **TFbind8.** The generated objects are DNA sequences with 8 nucleotides. The reward function is a binding affinity to a human transcription factor (Barrera et al., 2016), which is obtained via a pre-trained proxy model provided by Trabucco et al. (2022). We use a reward exponent of 3. We use a pre-defined set of modes provided by Shen et al. (2023).
- **L14-RNA1.** The generated objects are RNA sequences of length 14. The reward function is a binding affinity to a human transcription factor, which is obtained via a pre-trained proxy model from Sinai et al. (2020). We use a reward exponent of 8. We define modes as the top 0.01% quantile of $R(x)$, with diversity filtering whose threshold is 1 unit of Levenstein distance, also following Kim et al. (2024c).

Detailed training and hyperparameter settings for each task can be found in Appendix D.3.

**Baselines.** We compare our method with on-policy TB, $\epsilon$-exploration, and the PRT replay buffer method designed for biochemical tasks (Shen et al., 2023). Additionally, we evaluate it against a loss-prioritized replay buffer (PER) (Schaul, 2016). Since the local search method (Kim et al., 2024d) targets reward exploitation with a different purpose to ours, a separate analysis of its complementarity to our method is presented in Appendix E.5.

**Results.** As shown in Fig. 5, the Teacher method improves mode discovery when combined with both PER and PRT buffers, outperforming on-policy methods on every task. Similar trends are observed across other metrics, with faster convergence under the Teacher method. For TFbind8, our method's dominance is particularly evident for mode coverage metrics like EUBO and the number of modes, although ELBO remains comparable to using PER or PRT. In the largest task, L14-RNA1, the Teacher method surpasses PER and PRT in ELBO, EUBO, and the number of modes.

We draw special attention to the comparison between PER and PRT. While the differences are minimal in the first three tasks (QM9, TFbind8, sEH), in the larger-scale tasks, loss-prioritizing with PER shows a clear advantage. This suggests that loss information is more important for exploration in large-scale tasks, where the Teacher method with PER achieves the best overall performance.

## 6 DISCUSSION

We have introduced a **Teacher** that adaptively generates states for training a **Student** amortized sampler. The Teacher favors states for which the Student has high loss and thus promotes the discovery of new modes.

This approach paves the way for numerous future research directions. For example, an adaptive Teacher could be applied to amortize intractable inference in large language models (LLMs) (Hu et al., 2024) and diffusion models (Venkatraman et al., 2024), or to enhance exploration in automatic red-teaming for LLMs (Lee et al., 2024). Applying our method to probabilistic models such as amortized Bayesian causal discovery (Deleu et al., 2022; 2023; Nishikawa-Toomey et al., 2022) and amortized inference in graphical models (Falet et al., 2024) are also promising directions. Methodologically, the concept of using amortized prioritized experience replay (PER) as a Teacher can be extended to train general agent-based systems, not only amortized samplers.

ACKNOWLEDGEMENT

We thank to Anirudh Goyal, David Dobre, Gauthier Gidel for helpful discussion for this project. This research is based on funding from Samsung, Intel, CIFAR and the CIFAR AI Chair program. The research was enabled in part by computational resources provided by the Digital Research Alliance of Canada (https://alliancecan.ca), Mila (https://mila.quebec), and NVIDIA. This work was also partially supported by the National Research Foundation of Korea (NRF) grant funded by the Korea government (MSIT) (No. RS-2024-00410082).

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

## A    LIMITATIONS

The primary limitation of the adaptive Teacher is the added complexity in training, as the Teacher's policy network must be trained in addition to the Student's. This introduces a trade-off between increased training complexity and enhanced mode-seeking capabilities. While we believe that the ability to discover multiple modes outweighs the additional complexity, it is important to apply this technique judiciously. For tasks where the reward is unimodal or that do not require extensive exploration, using a Teacher may not be necessary. However, in scenarios where the model tends to collapse to specific modes of the multimodal target distribution, employing a Teacher is a beneficial choice.

Additionally, since the Student struggles to cover entire modes on its own, in significantly larger search spaces the Teacher may also have difficulty covering the full range of modes that the Student fails to discover, potentially collapsing into specific modes. In such large-scale settings, we expect that a multi-agent Teacher system—with multiple agents collaboratively covering the space—could be a beneficial direction for future work to mitigate this limitation.

## B    THEORETICAL ANALYSIS OF STATIONARY DISTRIBUTIONS

**Proposition 1.** *Let the behavior policy $P_\beta(\tau)$ be a distribution over trajectories $\tau \in \mathcal{T}$ that satisfies full support.*

*If the parameters $\theta^*$ and $\phi^*$ of the Student and Teacher policies, respectively, jointly optimize the objective functions to 0 in expectation over $P_\beta(\tau)$, then:*

*(a)  The marginal distribution of the Student policy over terminal states satisfies*

$$P_F^\top(x; \theta^*) \propto R(x),$$

*(b)  The marginal distribution of the Teacher policy over terminal states satisfies*

$$P_F^\top(x; \phi^*) \propto R(x)^\alpha,$$

*where:*

- *$R(x)$ is the reward function,*
- *$\epsilon > 0$ is an offset constant introduced in (5),*
- *$\alpha > 0$ is the mixing constant for the Teacher introduced in (6).*

*Proof.* According to the trajectory balance training theorem (Malkin et al., 2022), the Student policy achieves optimal loss (*i.e.*, $\mathcal{L}_{\text{TB}}(\tau; \theta^*) = 0$ for all $\tau$) if and only if its marginal distribution over terminal states $p(x, \theta^*) \propto R(x)$.

Suppose now that the Student policy has reached this optimum. Then the reward for the Teacher is

$$
\begin{aligned}
\log \tilde{R}_{\text{Teacher}}(x; \theta^*) &= \mathbb{E}_{P_B(\tau|x;\theta^*)} \left[ \log \left( \epsilon + \left( 1 + C \cdot \mathbb{I}_{\delta(\tau;\theta^*)>0} \right) \delta(\tau; \theta^*)^2 \right) \right] + \alpha \log R(x) \\
&= \mathbb{E}_{P_B(\tau|x;\theta^*)} \left[ \log(\epsilon) \right] + \alpha \log R(x) \\
&= \log(\epsilon) + \alpha \log R(x) \\
&= \log(\epsilon R(x)^\alpha).
\end{aligned}
\tag{10}
$$

Again by the TB training theorem, the loss is minimized if and only if the marginal distribution of the Student policy over terminal states is proportional to $\epsilon R(\cdot)^\alpha$, equivalently, to $R(\cdot)^\alpha$.

$\square$

## C  DETAILED IMPLEMENTATION

For all tasks, we set $\alpha = 0.5$ except for the exploration-intensive deceptive grid world tasks, where we use $\alpha = 0.0$. $C$ is set to 19 for all tasks. Our hyperparameter analysis is provided in Appendix E.4. In a diffusion sampling task, to concentrate the teacher on high-reward regions within a vast continuous space, we limit the teacher's training set to samples where $x > r_{\text{threshold}}$. Here, $r_{\text{threshold}}$ is the 90th percentile reward from the untrained student's samples.

For neural networks, we use identical architectures for both the Student and Teacher models. Specifically, for the GFN architecture design, we match the architectures used by each baseline for every task. Detailed descriptions are provided in Appendix D.

Regarding the replay buffer, we adhere to existing implementations for each task and follow prioritized replay rules. In grid world and diffusion sampling tasks, we use rank-based priority as introduced by Tripp et al. (2020) and Sendera et al. (2024). For biochemical tasks, we employ portion-wise priority as proposed by Shen et al. (2023). Additionally, we implemented two variants of a loss-prioritized buffer: one prioritizes based on TB loss, and the other uses $R_{\text{Teacher}}$ as the priority, serving a similar purpose. We analyze the performance differences between these variants and refer to the use of $R_{\text{Teacher}}$ as "PER."

When selecting the behavior policy during training, we periodically choose among the Student, Teacher, and buffer in specific proportions: a ratio of 1:1:0 for Grid World tasks, 3:1:2 for Diffusion Sampler tasks, and 2:1:3 for Biochemical tasks. In the diffusion sampler, we use PER for the replay buffer. For the biochemical task, we benchmark the teacher using both PER (referred to as PER + Teacher) and a reward-prioritized buffer (referred to as PRT + Teacher).

A higher Student ratio encourages exploitation, a higher Teacher ratio promotes exploration, and a larger buffer enhances sample efficiency until reaching the *replay ratio barrier* (D'Oro et al., 2023).

In deceptive grid world tasks, exploration is crucial, so we assign a higher Teacher ratio. For diffusion sampling tasks, using a prioritized replay buffer with Teacher rewards yields strong performance. Blending this approach slightly with the Teacher enhances both performance and convergence speed, achieving mode coverage for both EUBO and ELBO. In biochemical tasks, sample efficiency is critical. Therefore, existing tasks are set to be on-policy with a buffer ratio of 1:1. We adjust the proportions to favor on-policy learning by setting the Student-to-teacher ratio to 2:1, as biochemical tasks require some level of exploitation, as reported by Shen et al. (2023).

Other implementation details and hyperparameters are maintained as in each task's experimental setting.

## D  DETAILED EXPERIMENTAL SETTING

### D.1  DECEPTIVE GRID WORLD

**Evaluation metrics.** Following Bengio et al. (2021), we use the number of modes discovered and the empirical $L_1$ distance from target distribution as evaluation metrics. We define the *modes* as $x$'s with $R(x) = R_2 + R_0$. The exact size of the search space $|\mathcal{X}|$ is $(H-1)^d + d(H-1)$, with $d$ and $H$ representing the dimension and horizon of the hypergrid, respectively. $|\mathcal{X}|$ and the total number of modes $|\mathcal{M}|$ for each grid setting are reported in Table 3. The $L_1$ distance is calculated by $\frac{1}{|\mathcal{X}|} \sum_{x \in \mathcal{X}} [|p_\theta(x) - R(x)/Z|]$, where $Z = \sum_x R(x)$, which is known in this synthetic task. Unlike previous works (Bengio et al., 2021; Malkin et al., 2022) where a

Table 3: The total number of terminal states ($|\mathcal{X}|$) and modes ($|\mathcal{M}|$) for each grid setting, where $d$ is the dimension and $H$ is the horizon of the hypergrid.

|  | $|\mathcal{X}|$ | $|\mathcal{M}|$ |
|---|---|---|
| $d=2, H=128$ | 16383 | 676 |
| $d=2, H=256$ | 65535 | 2601 |
| $d=4, H=16$ | 64125 | 81 |
| $d=4, H=32$ | 1042685 | 1296 |

portion of the final training samples was used to approximate the expectation, we generate $10^5$ new samples from policy to calculate $p_\theta(x)$ for evaluation. We use one sample for only one gradient step by default, *i.e.*, update-to-data (UTD) ratio (Chen et al., 2021) is 1, except for the case using replay buffer, where the ratio increases to 2.

**Hyperparameters.** Following Bengio et al. (2021); Malkin et al. (2022), we use a two-layer MLP with 256 hidden units for the parameterized policy $P_F(\cdot; \theta)$ along with a learnable parameter for $\log Z_\theta$. We train them using the Adam optimizer with a learning rate of $10^{-3}$ for policy and $10^{-1}$ for $\log Z_\theta$. The backward policy $P_B$ is fixed as a uniform random policy. When using a replay buffer,

its size is dynamically set to $\lfloor 0.1|\mathcal{X}|\rfloor$. The total reward call budget is capped at 96,000, except for $(d = 4, H = 32)$, which is increased to 384,000, considering the significantly larger search space. We use a batch size of 16. The total number of gradient steps equals the number of reward calls divided by the batch size, and this number doubles when the replay buffer is used.

**Baseline implementations.** In our experiments, we set $\epsilon$ for $\epsilon$-exploration to 0.01, as this value yielded the lowest $L_1$ error among the tested options $\{0.001, 0.003, 0.01, 0.03, 0.1\}$ in the setting where $d = 2$ and $H = 128$. Similarly, for GAFN (Pan et al., 2023b), we set the intrinsic reward scale to 0.01, which also resulted in the lowest $L_1$ error among the values $\{1.0, 0.1, 0.01, 0.001\}$ under the same conditions. For PRT and PER, we use the same buffer size and prioritization scheme as above. Note that, unlike the original PER for value-based RL, the buffer we used contains only a terminal $x$ rather than all state transitions. From a sampled $x$, the trajectory can be constructed by backward generation using $P_B$. We also explore the integration of a transition-based replay buffer with the detailed balance objective in Appendix F.

### D.2 DIFFUSION SAMPLING

In our diffusion sampler, we primarily adhere to the settings of Sendera et al. (2024), which in turn build upon those of Zhang & Chen (2022).

**Tasks.** We benchmark the following two tasks:

*Gaussian Mixture Model with 25 Modes (25GMM).* The 25GMM consists of a two-dimensional Gaussian mixture with 25 modes, each having a variance of 0.3. The mode centers are positioned on the grid $\{-10, -5, 0, 5, 10\} \times \{-10, -5, 0, 5, 10\}$.

*Manywell (Noé et al., 2019).* The Manywell is a 32-dimensional distribution formed as the product of 16 identical two-dimensional double-well distributions. Each component is defined by the potential function

$$\mu(x_1, x_2) = \exp\left(-x_1^4 + 6x_1^2 + 0.5x_1 - 0.5x_2^2\right). \tag{11}$$

**Evaluation metrics.** We measure evidence lower bound (ELBO), importance sampled ELBO (ELBO-IS), and evidence upper bound (EUBO).

For estimating ELBO, we draw $M$ samples from current policy and take average value of estimated $\log Z$, which is $\log R(\cdot) + \log P_B(\cdot) - \log P_F(\cdot)$ as follows:

$$\text{ELBO} \approx \frac{1}{M} \sum_{i=1}^{M} (\log R(x_1^i) + \log P_B(\tau^i|x_1^i) - \log P_F(\tau^i; \theta)), \quad \tau^i \sim P_F(\tau; \theta), \tau^i \rightsquigarrow x_1^i, \tag{12}$$

where $\tau^i \rightsquigarrow x_1^i$ means that $x_1^i$ is the final state of $\tau^i$.

Calculation of ELBO-IS is similar to ELBO:

$$\text{ELBO-IS} \approx \log \frac{1}{M} \sum_{i=1}^{M} \exp(\log R(x_1^i) + \log P_B(\tau^i|x_1^i) - \log P_F(\tau^i; \theta)), \quad \tau^i \sim P_F(\tau; \theta), \tau^i \rightsquigarrow x_1^i. \tag{13}$$

EUBO was introduced as a metric that measures mode coverage by Blessing et al. (2024). To calculate EUBO, we sample $M$ samples from the target distribution and take the average of their variational log-likelihood bounds as follows:

$$\text{EUBO} \approx \frac{1}{M} \sum_{i=1}^{M} (\log R(x_1^i) + \log P_B(\tau^i|x_1^i) - \log P_F(\tau^i; \theta)) \quad x_1^i \sim P^*(x_1), \tau^i \sim P_B(\tau \mid x_1^i). \tag{14}$$

**Forward and backward transition modeling.** The diffusion sampler models discretized SDE trajectories $\tau = (x_0 \rightarrow x_{\Delta t} \rightarrow \ldots \rightarrow x_1)$, starting from $x_0 = (\mathbf{0}, t = 0)$. Here, $\Delta t = 1/T$, where $T$ is the number of discrete time steps.

The forward policy $P_F(x_{t+\Delta t} \mid x_t; \theta)$ is modeled as a Gaussian distribution with mean $x_t + u(x_t, t; \theta)\Delta t$ and covariance $\sigma^2 \Delta t \, \mathbb{I}$:

$$P_F(x_{t+\Delta t} \mid x_t; \theta) = \mathcal{N}\left(x_{t+\Delta t}; x_t + u(x_t, t; \theta)\Delta t, \sigma^2 \Delta t \, \mathbb{I}\right). \tag{15}$$

Here, $u(x_t, t; \theta)$ is the learnable score function, $\sigma$ is the standard deviation, and $\mathbb{I}$ denotes the identity matrix to ensure isotropic covariance.

The backward policy $P_B(x_{t-\Delta t} \mid x_t)$ is defined as a discretized Brownian bridge with noise rate $\sigma$:

$$P_B(x_{t-\Delta t} \mid x_t) = \mathcal{N}\left(x_{t-\Delta t}; \frac{t-\Delta t}{t}x_t, \frac{t-\Delta t}{t}\sigma^2 \Delta t \, \mathbb{I}\right). \tag{16}$$

The densities of the distributions over complete forward and backward trajectories are given by:

$$P_F(\tau; \theta) = \prod_{i=0}^{T-1} P_F(x_{(i+1)\Delta t} \mid x_{i\Delta t}; \theta), \quad P_B(\tau \mid x_1) = \prod_{i=1}^{T-1} P_B(x_{i\Delta t} \mid x_{(i+1)\Delta t}). \tag{17}$$

The diffusion policy parameter $\theta$ is trained using the TB loss.

**Hyperparameters.** We set $\sigma^2 = 5.0$ for 25GMM and $\sigma = 1.0$ for Manywell, with the number of time steps $T = 100$, following Sendera et al. (2024). We employ the same architecture as Zhang & Chen (2022) and Sendera et al. (2024), increasing the hidden dimension from 64 to 256 for Manywell to accommodate the 32-dimensional tasks, and apply this adjustment to all baselines. For replay buffer capacity, we set it to 5,000 for the 25GMM task and 20,000 for the Manywell task. All learning hyperparameters remain identical to those in Sendera et al. (2024). For evaluation, we set the number of samples to $M = 2,000$.

## D.3    BIOLOGICAL AND CHEMICAL DISCOVERY

For biochemical design tasks, we mostly follow the setting of Shen et al. (2023). For all tasks, we use a prepend-append MDP (PA-MDP), where the action is defined as adding a token at the beginning or the end of a partial sequence. This MDP formulation makes it possible to have multiple trajectories associated with the same design configuration $x$.

**Hyperparameters.** For training GFlowNets, we use similar setting proposed by prior works (Shen et al., 2023; Kim et al., 2024d). We use Adam optimizer (Kingma & Ba, 2015) with learning rate $10^{-2}$ for $\log Z_\theta$, $10^{-4}$ for forward policy, $5 \times 10^{-4}$ for teacher policy. To parametrize forward policy, we adopt relative edge flow policy parametrization mapping (SSR) from Shen et al. (2023). For QM9 and sEH tasks, we employ a two-layer architecture with 1024 hidden units, while for the other tasks, we choose to use a two-layer architecture with 128 hidden units. We initialize $\log Z_\theta$ to 5.0 for all methods. For backward policy, we use a fixed uniform policy. In terms of reward exponent, we use a value of 20 for both QM9 and TFbind8. For sEH and L14-RNA1, we use relatively higher values, 200 and 40, respectively.

**Evaluation metrics.** For evaluation, we compute the number of modes using all the samples collected over the course of training. What we count as a mode should have a high reward. Unlike with other designs, we define "mode" as a configuration whose reward is above a certain threshold. This is different from previously used metrics of mode counting to assess diversity. For QM9 and TFbind8, we use a default mode set suggested by (Shen et al., 2023). For sEH, we set the reward threshold as the top 0.01% of $\mathcal{X}$ in terms of the reward and the diversity threshold as 0.4 Tanimoto diversity. For L14-RNA1, we set the reward threshold as the top 0.01% of $\mathcal{X}$ in terms of the reward and the diversity threshold as 1 unit of Levenstein distance. We also report ELBO and EUBO which are described in Appendix D.2. We generate $M = 2,048$ samples from both the trained policy and the target distribution for the ELBO and EUBO calculation. To sample from the target distribution, we evaluate all possible configurations and sample them proportionally to their reward. As the number of possible configurations is enormously large for sEH and L14-RNA1 tasks, we use the Gumbel-max trick (Gumbel, 1954) for sampling from the discrete probability distribution.

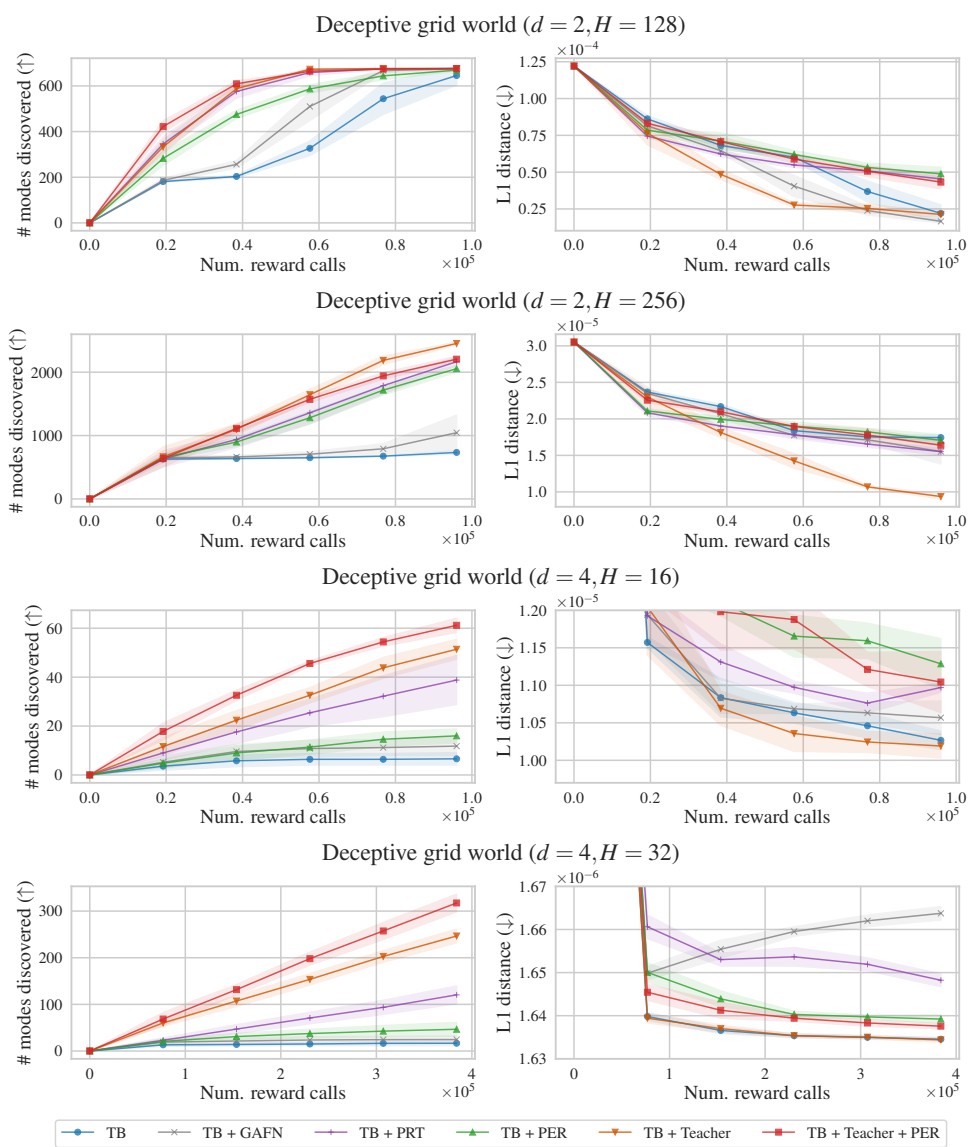

Figure 6: Evolution of evaluation metrics for each method in deceptive grid world task. The mean value with standard deviation is depicted from five independent runs.

# E    ADDITIONAL EXPERIMENTAL RESULTS

## E.1    EXTENDED EXPERIMENTAL RESULTS OF DECEPTIVE GRID WORLD

Fig. 6 shows the changes in two evaluation metrics during training: the number of modes discovered and the empirical $L_1$ distance between the target and sampled distributions. The results indicate that Teacher is highly effective at discovering modes and also generally performs well in terms of the $L_1$ distance. In contrast, while PER and PRT improve mode discovery, they tend to slightly worsen $L_1$ performance. We believe this is because the replay buffer provides a regularizing signal that prevents mode collapse, making the policy learning process more challenging compared to using purely on-policy methods without regularization. This leads to slight underfitting. On the other hand, on-policy methods can easily overfit to the target distribution (but into specific modes), resulting in descent $L_1$ performance. However, without off-policy regularization, they tend to drop modes and fail to cover all modes in the target distribution. Teacher achieves high performance in both $L_1$ and mode coverage, indicating that it not only offers off-policy regularization to ensure comprehensive mode coverage but also provides efficient curricula for faster convergence to the target distribution, leading to strong performance across both metrics.

Table 4: The effect of number of Monte Carlo samples ($N_{\mathrm{MC}}$) in deceptive grid world task.

| Grid config. $\rightarrow$ | $d = 2, H = 256$ | | $d = 4, H = 32$ | |
|---|---|---|---|---|
| Algorithm $\downarrow$ Metric $\rightarrow$ | # modes ($\uparrow$) | $L_1 \times 10^{-5}$ ($\downarrow$) | # modes ($\uparrow$) | $L_1 \times 10^{-6}$ ($\downarrow$) |
| $N_{\mathrm{MC}} = 1$ (default) | $2452.6 \pm 21.7$ | $\mathbf{0.94} \pm 0.03$ | $\mathbf{246.6} \pm 14.7$ | $\mathbf{1.634} \pm 0.001$ |
| $N_{\mathrm{MC}} = 3$ | $2489.6 \pm 15.8$ | $0.96 \pm 0.07$ | $234.6 \pm 14.2$ | $1.635 \pm 0.000$ |
| $N_{\mathrm{MC}} = 5$ | $2490.4 \pm 30.3$ | $\mathbf{0.94} \pm 0.06$ | $230.8 \pm 3.4$ | $\mathbf{1.634} \pm 0.000$ |
| $N_{\mathrm{MC}} = 10$ | $\mathbf{2492.0} \pm 26.3$ | $0.96 \pm 0.05$ | $239.0 \pm 11.8$ | $\mathbf{1.634} \pm 0.000$ |

Table 5: Ablation study on $C$ in deceptive grid world task.

| Grid config. $\rightarrow$ | $d = 2, H = 256$ | | $d = 4, H = 32$ | |
|---|---|---|---|---|
| Algorithm $\downarrow$ Metric $\rightarrow$ | # modes ($\uparrow$) | $L_1 \times 10^{-5}$ ($\downarrow$) | # modes ($\uparrow$) | $L_1 \times 10^{-6}$ ($\downarrow$) |
| $C = 0$ | $\mathbf{2469.4} \pm 29.7$ | $1.06 \pm 0.08$ | $253.2 \pm 11.3$ | $\mathbf{1.634} \pm 0.000$ |
| $C = 9$ | $2454.2 \pm 18.7$ | $0.95 \pm 0.02$ | $\mathbf{256.0} \pm 15.4$ | $1.635 \pm 0.000$ |
| $C = 19$ (default) | $2452.6 \pm 21.7$ | $\mathbf{0.94} \pm 0.03$ | $246.6 \pm 14.7$ | $\mathbf{1.634} \pm 0.001$ |
| $C = 29$ | $2465.2 \pm 30.6$ | $\mathbf{0.94} \pm 0.04$ | $243.4 \pm 8.5$ | $\mathbf{1.634} \pm 0.000$ |

### E.2 STUDY ON MONTE CARLO APPROXIMATION FOR $R_{\mathrm{TEACHER}}$

We use Monte Carlo estimate with a single trajectory to approximate the expectation in Eq. (4) and Eq. (5). To validate that this single-sample approximation is reasonable, we test our algorithm in the deceptive grid world task with an increased number of samples, ranging from 3 to 10. We do not use the replay buffer in this analysis. As described in Appendix E.1, the performance is not significantly affected by $N_{\mathrm{MC}}$. This suggests that using a Monte Carlo approximation with a sample size of 1 to estimate the stochastic reward $R_{\mathrm{Teacher}}$ was reasonable.

### E.3 ABLATION STUDY ON THE CHOICE OF $C$ VALUE

We set the hyperparameter $C$ in Eq. (5) to 19 across all experiments without an extensive hyperparameter search. To evaluate this choice, we perform an ablation study on the deceptive grid world task, testing alternative values of 0, 9, and 29. Note that the replay buffer is not used for this analysis. The results are summarized in Table 5. Although the effect on the number of modes discovered is somewhat mixed, $C = 0$ performs slightly worse than the other values regarding the empirical $L_1$ error, supporting our hypothesis that focusing more on undersampled regions is beneficial. We also found that the results are not highly sensitive to the choice of $C$.

### E.4 STUDY ON $\alpha$ OF REWARD MIXING

We introduce $\alpha$ to mix the reward based on the Student's loss and Student's log reward to help Teacher target both high-loss and high-reward areas effectively. In this section, we investigate the effect of $\alpha$ on the performance of our method.

Table 6: Mixing component study on deceptive grid world task

| Grid config. $\rightarrow$ | $d = 2, H = 256$ | | $d = 4, H = 32$ | |
|---|---|---|---|---|
| Algorithm $\downarrow$ Metric $\rightarrow$ | # modes ($\uparrow$) | $L_1 \times 10^{-5}$ ($\downarrow$) | # modes ($\uparrow$) | $L_1 \times 10^{-6}$ ($\downarrow$) |
| $\alpha = 0.0$ | $2452.6 \pm 21.7$ | $0.94 \pm 0.03$ | $246.6 \pm 14.7$ | $1.634 \pm 0.001$ |
| $\alpha = 0.5$ | $2415.2 \pm 262.8$ | $0.90 \pm 0.11$ | $85.6 \pm 8.5$ | $1.634 \pm 0.000$ |

**Deceptive grid world.** Table 6 shows that reward mixing with $\alpha = 0.5$ degrades performances for mode seeking in high dimensional tasks as deceptive grid world task is exploration intensive task; teacher solely focusing on high loss region is more beneficial. Still mixing with $\alpha = 0.5$ outperforms other baselines.

Table 7: Mixing component study on diffusion sampler task

| Energy $\rightarrow$ | 25GMM ($d = 2, \log Z = 0$) | | | Manywell ($d = 32, \log Z = 164.696$) | | |
|---|---|---|---|---|---|---|
| Algorithm $\downarrow$ Metric $\rightarrow$ | ELBO ($\uparrow$) | ELBO-IS ($\uparrow$) | EUBO ($\downarrow$) | ELBO ($\uparrow$) | ELBO-IS ($\uparrow$) | EUBO ($\downarrow$) |
| $\alpha = 0.0$ | $-0.144_{\pm 0.001}$ | $-0.009_{\pm 0.006}$ | $0.122_{\pm 0.010}$ | $163.447_{\pm 0.063}$ | $164.694_{\pm 0.060}$ | $166.024_{\pm 0.001}$ |
| $\alpha = 0.5$ | $-0.137_{\pm 0.004}$ | $-0.005_{\pm 0.007}$ | $0.115_{\pm 0.009}$ | $163.484_{\pm 0.049}$ | $164.676_{\pm 0.048}$ | $165.800_{\pm 0.045}$ |

**Diffusion sampling.** As shown in Table 7, mixing with $\alpha = 0.5$ shows slightly better performance, though both achieve significantly higher sampling efficiency compared to the baselines. Both $\alpha = 0.0$ and $\alpha = 0.5$ come close to reaching the target $\log Z$.

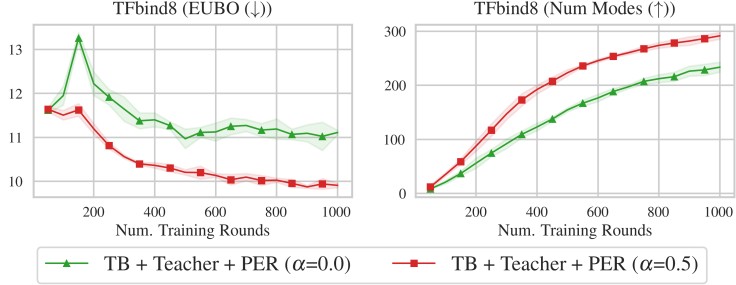

Figure 7: Training graph for TFbind8 task by varying $\alpha$ of reward mixing. Mean value with standard deviation is depicted five independent runs.

**Biological and Chemical Discovery (TFbind8).** As shown in Fig. 7, mixing with $\alpha = 0.5$ yields significantly better performance in TFbind8 task. This highlights the importance of having the teacher focus on both high-loss and high-reward areas.

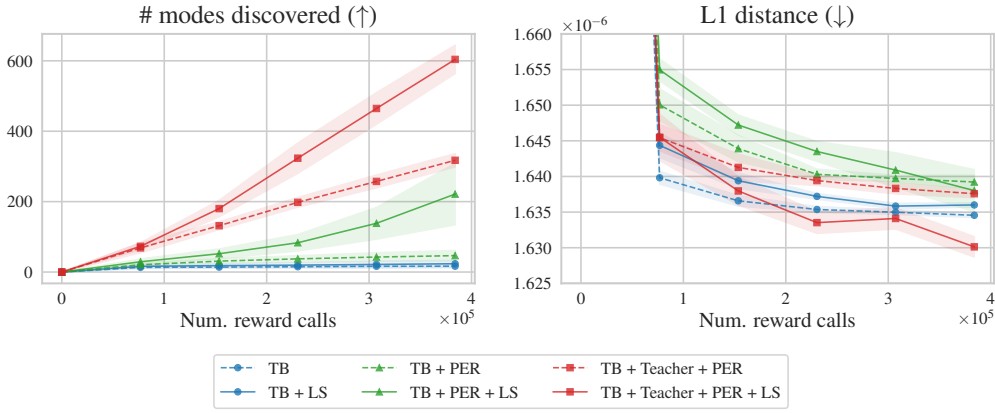

Figure 8: Evolution of evaluation metrics with or without the local search in deceptive grid worlds ($d = 4, H = 32$). The mean value with standard deviation is depicted from five independent runs.

Table 8: Comparison with local search (LS) (Sendera et al., 2024) on Manywell

| Energy → | Manywell ($d = 32$, $\log Z = 164.696$) | | |
|---|---|---|---|
| Algorithm ↓ | ELBO (↑) | ELBO-IS (↑) | EUBO (↓) |
| PER | $161.537_{\pm 0.186}$ | $162.582_{\pm 0.268}$ | $210.440_{\pm 6.888}$ |
| PER + LS | $163.308_{\pm 0.189}$ | $164.508_{\pm 0.168}$ | $168.953_{\pm 3.209}$ |
| Teacher | $\mathbf{163.484}_{\pm 0.049}$ | $164.676_{\pm 0.048}$ | $165.800_{\pm 0.045}$ |
| Teacher + LS | $163.472_{\pm 0.086}$ | $\mathbf{164.685}_{\pm 0.063}$ | $\mathbf{165.787}_{\pm 0.044}$ |

## E.5 TEACHER WITH LOCAL SEARCH

Local search is a useful technique to improve the sampling quality of GFlowNets (Hu et al., 2023; Kim et al., 2024d). In this section, we investigate the possible integration of local search and Teacher and compare it with existing local search integrated solely on Student.

**Deceptive grid world.** We tested the backtracking-and-reconstruction local search method introduced in §3.3 in the deceptive grid world, using grid configurations of ($d = 2, H = 256$) and ($d = 4, H = 32$). Four iterative local searches were performed every 16th training batch. The backtracking ratio is 0.5, meaning the last half of a trajectory is destroyed and reconstructed by policy. We applied a deterministic acceptance rule, accepting a new trajectory if it had a higher $R_{\text{Teacher}}$. For comparison, we used two baselines: on-policy TB and TB with PER, both using the same local search but using the task reward $R$ to determine the acceptance.

The experimental results extending is illustrated in Fig. 8. Teacher with PER outperforms both PER and on-policy TB with a large margin in terms of mode coverage, regardless of whether local search is applied. When combined with local search, Teacher achieves the best results in both the number of modes discovered and the empirical $L_1$ distance. We believe this performance gain is largely due to the reduction of non-stationarity §3.3, though isolating the exact contribution is complex and left for future work.

**Diffusion sampling.** We utilize the Manywell task to compare the effect of local search on the Teacher model. Specifically, we employ parallel local search methods (Sendera et al., 2024) that leverage Metropolis-Hastings-guided Langevin dynamics (MALA) on samples from the replay buffer to refine sample quality. As shown in the Table 8, by integrating this local search with Prioritized Experience Replay (PER), we observe significant performance improvements.

Remarkably, the Teacher model—even without local search—still outperforms these results. This is notable because the Teacher's exploration does not require gradient information of the energy function, whereas MALA relies on such gradient information. Since the Teacher rapidly achieves optimal sampling quality on the Manywell task, we observe not much improvement when applying local search to the Teacher in the diffusion sampling task.

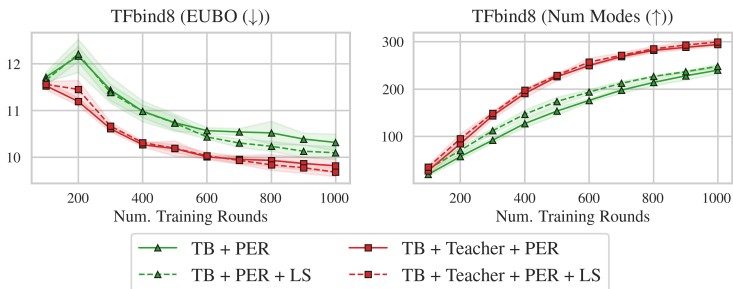

Figure 9: Training graph for TFbind8 task by integrating local search (Kim et al., 2024d). Mean value with standard deviation is depicted five independent runs.

**Biological and Chemical Discovery.** We utilize the TFbind8 task to compare the effect of local search on the Teacher model. Specifically, we employ the local search method suggested by (Kim et al., 2024d), which involves backtracking and reconstructing sequences using the forward and backward policies of GFlowNets. The decision to accept adjusted samples is based on whether $R(x') > R(x)$, where $x'$ is the new sample. For the Teacher model, as described in §3.3, we perform local search to mitigate non-stationary in the student and to optimize $R_{\text{teacher}}(x)$. For both the Student and Teacher models, we employ Prioritized Experience Replay (PER).

We visualize the results in Fig. 9. As shown in the figure, the Teacher model without local search still outperforms the Student model with local search. This demonstrates that the exploration capability of the Teacher model is far more efficient than conducting local search with the current policy. Moreover, we observe that integrating local search with the Teacher model leads to further improvement in terms of EUBO, highlighting the synergistic effect of combining the Teacher model with local search.

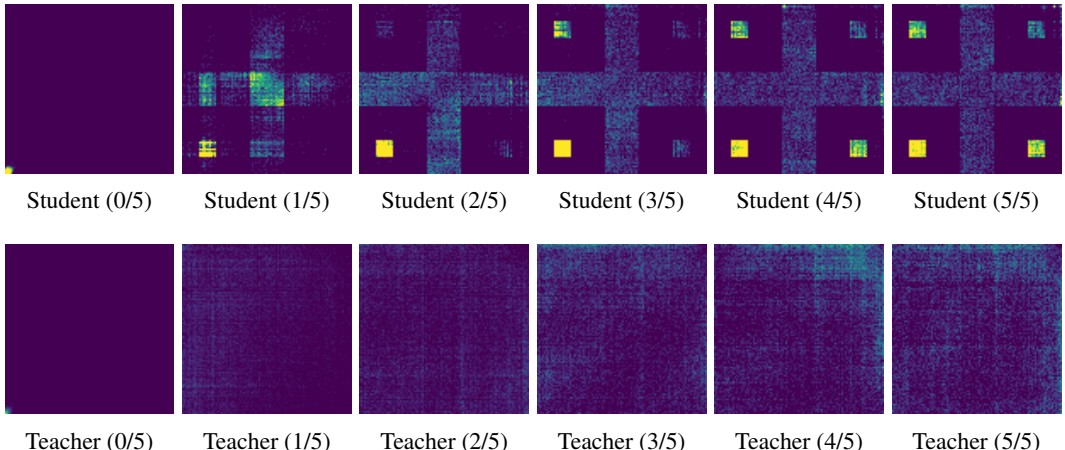

Figure 10: Illustration of the distribution dynamics between the Teacher and Student models, along with their stationary distributions. The Student (*ratio*) represents the fraction of completed training steps.

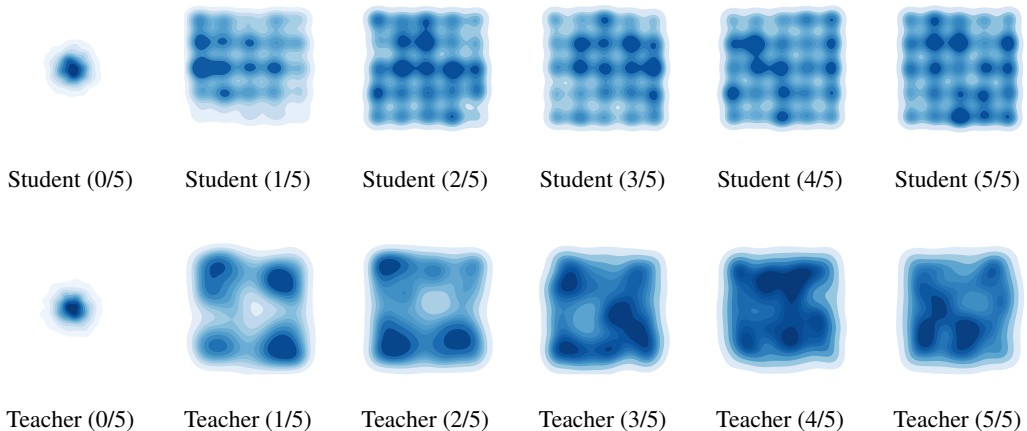

Figure 11: Illustration of the distribution dynamics between the Teacher and Student models, along with their stationary distributions. The Student (*ratio*) represents the fraction of completed training steps.

## E.6  2D PLOTS OF TEACHER AND STUDENT OVER TRAINING

This section presents the distributions of the Teacher and Student models during training. For this visualization, we set the mixing component $\alpha = 0$, meaning the Teacher's major objective is to explore the Student's loss regions. As shown in Fig. 10 and Fig. 11, the Teacher effectively identifies the Student's missing modes, providing a suitable training distribution throughout the epochs, ultimately enabling the Student to discover all the modes successfully.

Table 9: Evaluation results of DB algorithms on deceptive grid worlds with dimension $d$ and grid length $H$. Mean and standard deviation from 5 independent runs are reported. The **bold** is applied to the best mean value among DB-based methods.

| Grid config. $\rightarrow$ | $d = 2, H = 128$ | | $d = 2, H = 256$ | | $d = 4, H = 16$ | | $d = 4, H = 32$ | |
|---|---|---|---|---|---|---|---|---|
| Algorithm $\downarrow$ Metric $\rightarrow$ | # modes ($\uparrow$) | $L_1 \times 10^{-5}$ ($\downarrow$) | # modes ($\uparrow$) | $L_1 \times 10^{-5}$ ($\downarrow$) | # modes ($\uparrow$) | $L_1 \times 10^{-5}$ ($\downarrow$) | # modes ($\uparrow$) | $L_1 \times 10^{-6}$ ($\downarrow$) |
| TB   (on-policy $\rightarrow$) | 645.4 ± 41.5 | 2.20 ± 0.58 | 733.6 ± 25.1 | 1.74 ± 0.04 | 6.6 ± 2.5 | 1.027 ± 0.012 | 16.6 ± 4.8 | 1.635 ± 0.000 |
| + Teacher | 676.0 ± 0.0 | 2.13 ± 0.18 | 2452.6 ± 21.7 | 0.94 ± 0.03 | 51.4 ± 4.0 | 1.019 ± 0.016 | 246.6 ± 14.7 | 1.634 ± 0.001 |
| DB   (on-policy $\rightarrow$) | 644.0 ± 13.3 | 4.57 ± 0.12 | 1025.4 ± 132.8 | 1.69 ± 0.03 | 1.8 ± 1.7 | **1.003** ± 0.009 | 6.8 ± 2.9 | 1.634 ± 0.000 |
| + $\epsilon$-expl. | 578.2 ± 25.9 | 5.23 ± 0.07 | 814.6 ± 23.4 | 1.75 ± 0.03 | 2.4 ± 0.8 | 1.010 ± 0.004 | 25.2 ± 2.6 | 1.635 ± 0.000 |
| + PER* | 316.6 ± 166.5 | 6.01 ± 0.62 | 899.4 ± 241.7 | 1.75 ± 0.03 | 2.6 ± 1.0 | 1.005 ± 0.012 | 17.2 ± 12.5 | 1.634 ± 0.000 |
| + Teacher | **675.4** ± 0.5 | **4.42** ± 0.15 | **1817.6** ± 49.8 | **1.59** ± 0.01 | **58.2** ± 5.1 | 1.009 ± 0.007 | **345.2** ± 41.2 | **1.632** ± 0.004 |

## F   TEACHER FOR DETAILED BALANCE

In this section, we extend the proposed idea in §3 to detailed balance (DB; Bengio et al., 2023), another GFlowNet learning objective.

### F.1   DETAILED BALANCE AND TEACHER'S REWARD FOR DB

The general problem settings, including MDP formulation and reward function, are the same as §2. Unlike TB, which requires parameterizing $Z_\theta$, DB parameterizes the *state flow function* $F(s;\theta)$ for each state, along with $P_F$ and $P_B$. Note that $F(x;\theta) = R(x)$ for every terminal state $x \in \mathcal{X}$, and the initial state flow $F(s_0)$ is an estimate of the total reward. The detailed balance discrepancy is defined for any transition of states $(s, a, s')$ as

$$\delta_{\text{DB}}(s, a, s'; \theta) := \underbrace{[\log F(s'; \theta) + \log P_B(s \mid s'; \theta)]}_{\text{backward edge flow}} - \underbrace{[\log F(s; \theta) + \log P_F(s' \mid s; \theta)]}_{\text{forward edge flow}}. \tag{18}$$

Same as TB, when $\delta_{\text{DB}}(s, a, s'; \theta) = 0$ for all $(s, a, s')$, then $P_F^\top(x) = R(x)/Z$ is achieved for all $x$, where $P_F^\top$ defined as Eq. (1). We can naturally define a DB loss as $\delta_{\text{DB}}(s, a, s'; \theta)^2$ on each transition $(s, a, s')$ sampled from a behavior policy $\pi$. For more formal derivation, please refer to Bengio et al. (2023).

Analogous to Eq. (4) and Eq. (5), we define the basic form as and the re-weighted version of Teacher's reward for DB. The basic form is

$$\log R_{\text{Teacher-DB}}^{\text{basic}}(x; \theta) = \mathbb{E}_{P_B(\tau|x)} \left[ \log \left( \sum_{(s,a,s') \in \tau} \delta_{\text{DB}}(s, a, s'; \theta)^2 \right) \right], \tag{19}$$

and the re-weighted version is

$$\log R_{\text{Teacher-DB}}^{\text{weighted}}(x; \theta) = \mathbb{E}_{P_B(\tau|x;\theta)} \left[ \log \left( \epsilon + \sum_{(s,a,s') \in \tau} \left( 1 + C \mathbb{I}_{\delta_{\text{DB}}(s,a,s';\theta) > 0} \right) \delta_{\text{DB}}(s, a, s'; \theta)^2 \right) \right], \tag{20}$$

where we approximate the expectation using a single trajectory.

The reward mixing in Eq. (6) is not directly available in the DB case since it entails a non-trivial credit assignment problem. Thus, we set $R_{\text{Teacher-DB}} = R_{\text{Teacher-DB}}^{\text{weighted}}$.

The overall training procedure is similar to §3.2 and Algorithm 1, except we use DB loss for both Student and Teacher training. For a given transition $(s, a, s') \sim \pi$, the DB-loss functions are defined by

$$\mathcal{L}_{\text{Student-DB}}(s, a, s'; \theta) = \delta_{\text{DB}}(s, a, s'; \theta)^2 = \left( \log \frac{F(s; \theta) P_F(s' \mid s; \theta)}{F(s'; \theta) P_B(s \mid s')} \right)^2, \tag{21}$$

$$\mathcal{L}_{\text{Teacher-DB}}(s, a, s'; \phi) = \delta_{\text{Teacher-DB}}(s, a, s'; \phi)^2 = \left( \log \frac{F(s; \phi) P_F(s' \mid s; \phi)}{F(s'; \phi) P_B(s \mid s')} \right)^2, \tag{22}$$

where $F(x; \theta) = R(x)$ and $F(x; \phi) = R_{\text{Teacher-DB}}$.

## F.2 Experiments in deceptive grid world

We use the same experimental settings as §5.1. We incorporate a transition-based replay buffer, meaning that we save all state transitions along the trajectory rather than saving only the terminal state $x$ as in the TB case. This allows a closer implementation of the original PER, where the prioritization is performed with a TD error for each transition. To distinguish the PER used in §5.1, we call the transition-based PER as PER*. Regarding the baselines, we do not benchmark PRT as it is not trivial to assign an episodic reward at the terminal state to each transition. We omit the GAFN since its source code only supports the TB algorithm. We also include TB, TB with Teacher for reference.

The result is summarized in Table 9. Similar to the TB case (Table 1), Teacher provides a significant improvement over baselines for DB. This confirms that our method offers flexibility across different GFlowNets objective functions.

## G Scaling experiments

In this section, we demonstrate the scalability of our method. We first test it on larger-scale tasks on Deceptive Gridworlds, to show that its effectiveness remains consistent as the scale increases. Then we apply our method to a real-world task of prompt sampling on large language models (LLMs), where we discover desirable prompt sentences that require effective exploration of the combinatorially large language search space.

Table 10: Evaluation results on large-scale deceptive grid worlds with dimension $d$ and grid length $H$. The mean and standard deviation of the number of modes discovered (# modes) from 3 independent runs are reported. Due to the computational expense of obtaining the exact target distribution in large-scale problems, $L_1$ distance is excluded from the analysis. The best mean values are highlighted in **bold**, while the second-best are marked with an underline.

| Grid config. $\rightarrow$ | $d = 4, H = 64$ | $d = 4, H = 128$ | $d = 6, H = 32$ | $d = 6, H = 64$ |
|---|---|---|---|---|
| Num. terminal states $|\mathcal{X}| \rightarrow$ | $1.68 \times 10^7$ | $2.68 \times 10^8$ | $1.06 \times 10^9$ | $6.85 \times 10^{10}$ |
| TB (on-policy $\rightarrow$) | $24.0 \pm 5.7$ | $228.7 \pm 38.1$ | $0.3 \pm 0.5$ | $3.7 \pm 2.6$ |
| + $\epsilon$-expl. | $49.7 \pm 17.2$ | $\mathbf{866.7} \pm 154.4$ | $0.7 \pm 0.5$ | $\underline{11.3} \pm 4.0$ |
| + GAFN | $42.0 \pm 6.5$ | $180.0 \pm 51.9$ | $1.3 \pm 1.2$ | $4.0 \pm 2.2$ |
| + PRT | $\underline{119.3} \pm 16.0$ | $222.7 \pm 11.6$ | $\underline{6.7} \pm 0.5$ | $7.3 \pm 0.9$ |
| + PER | $70.7 \pm 12.3$ | $164.3 \pm 23.7$ | $2.0 \pm 1.4$ | $5.3 \pm 0.9$ |
| + Teacher (*ours*) | $\mathbf{299.0} \pm 10.7$ | $\underline{728.0} \pm 192.0$ | $\mathbf{9.7} \pm 3.4$ | $\mathbf{21.3} \pm 6.0$ |

### G.1 Large-scale experiment on deceptive hypergrids

We evaluate our algorithm on larger-scale settings of deceptive grid worlds. Details of the grid configurations and the total number of terminal states (representing the size of the search space) are provided in Table 10. We use the same experimental settings we used for the grid with ($d = 4, H = 32$), which are described in Appendix D.1. Note that we omit $L_1$ distance from our metrics, as calculating the exact target distribution is computationally infeasible for these large-scale problems.

As shown in the results Table 10, our algorithm generally outperforms other baselines even in environments with a much larger search space. We conjecture the strong performance of $\epsilon$-exploration in the ($d = 4, H = 128$) configuration is because, when $H$ is large, clusters of adjacent high-reward states are large. In such cases, the random, brittle actions from $\epsilon$-exploration can be advantageous in discovering multiple adjacent modes within the same region.

### G.2 Sampling LLM attack prompts

We benchmark our off-policy training method on the automated red-teaming task using GFlowNets, following the approach of Lee et al. (2024). In this task, the problem is formulated as inference over prompt sequences proportional to a target reward, which is computed by a toxicity classifier evaluated on the response the prompt induces from a fixed victim model. We adhere to the same settings, baselines, evaluation model, and model architecture as the mentioned work. Specifically, we fine-tune GPT-2 as a GFlowNet policy to serve as the attack model.

**Setting.** The log-reward is defined as a weighted mixture of the log-likelihood of the language model and the toxicity score of the prompt. Toxicity is evaluated by a classifier that measures how likely the prompt is to induce toxic outputs from the victim model.

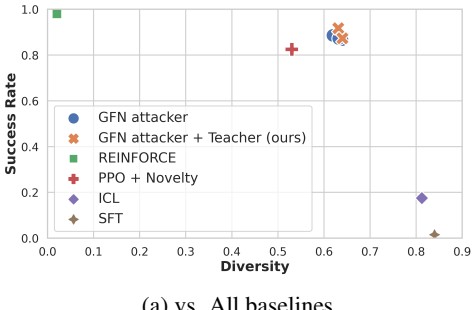
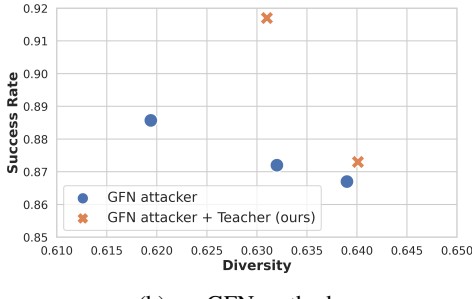

(a) vs. All baselines                          (b) vs. GFN methods

Figure 12: Percentage of toxic prompts (y-axis) versus prompt diversity (x-axis, measured as 1−cosine similarity) for attack methods on GPT-2.

**Baselines.** We compare our method with several baselines: In-Context Learning (ICL), Supervised Fine-Tuning (SFT), REINFORCE, and Proximal Policy Optimization (PPO) with a novelty reward as suggested by Hong et al. (2024). We also include the GFlowNet attacker proposed by Lee et al. (2024), which is the state-of-the-art method leveraging a GFlowNet sampler with sophisticated off-policy techniques using a replay buffer.

**Implementation.** We implement a teacher network over the GFlowNet attacker. While the original GFlowNet attacker uses on-policy updates and a 1:1 ratio in the replay buffer, we use Teacher, Student (on-policy), and replay buffer trajectories in a 2:1:3 ratio. We aim to observe whether this adjustment provides any benefits over the baseline. We reproduce the GFN attacker results with $\beta \in 0.06, 0.07, 0.08$, where $\beta$ is the temperature parameter for the toxicity reward. For our teacher method, we test with $\beta \in 0.02, 0.05$. For other baseline results, we directly use the data reported in the figures by Lee et al. (2024).

**Results.** As shown in Fig. 12a, the teacher network achieves slightly higher diversity and success rates compared to the state-of-the-art GFlowNet attacker. Other baselines fail to produce both diverse and toxic prompts: REINFORCE leads to mode collapse, and ICL and SFT do not generate meaningful toxic prompts (see Lee et al. (2024) for a more detailed analysis of these baselines). The GFlowNet attacker provides well-balanced results, achieving high toxicity in successful prompt sentences with diversity. Our Teacher network offers a slight improvement over the basleine GFlowNet attacker method (see Fig. 12b), demonstrating that our approach can be flexibly applied to real-world tasks. Notably, even when using a lower $\beta$ (indicating a peaky toxicity reward) than the GFlowNet attacker, the diversity achieved is higher. This suggests that the teacher encourages exploration into missing modes to enhance diversity.

These findings show the potential applicability of the teacher concept in large language model reasoning tasks where amortized inference using off-policy RL has been applied, including automated red-teaming, infilling, chain-of-thought reasoning, and planning (as studied in Hu et al., 2024; Song et al., 2024; Yu et al., 2024).

