# OpenReview forum: "Adaptive teachers for amortized samplers"
_ICLR.cc/2025/Conference — ICLR 2025 Poster_

### Official Review · Reviewer_CGLL · 2024-10-30

**Soundness:** 3
**Presentation:** 3
**Contribution:** 2
**Rating:** 5
**Confidence:** 4

**Summary:**

This paper introduces a method for training a neural network to approximate a distribution with a specified unnormalized density. Specifically, it proposes an adaptive training distribution, termed the "Teacher," which guides the primary amortized sampler, or "Student," by prioritizing high-loss regions.

**Strengths:**

The Teacher model has the potential to generalize across the Student's high-loss regions, thereby improving mode coverage. The algorithm’s effectiveness is demonstrated through both discrete and continuous tasks, comparing favorably against other GFlowNet algorithms.

**Weaknesses:**

This paper appears to fall slightly below the standard expected at ICLR for several reasons:

1. A critical issue is the lack of a solid theoretical foundation; the paper primarily reports numerical results without a deeper mathematical analysis. For instance, there is no mathematical description or guarantee of convergence rate for Algorithm 1.

2. Regarding the experiments, the explanation of the architecture design and the choice of hyperparameters would benefit from greater clarity and justification. I will outline these concerns in more detail in the questions below.

**Questions:**

1. Convergence Rate of Algorithm 1: What is the convergence rate of your Algorithm 1? Although it performs well in exploring more modes in the example tasks, the convergence rate is also a key factor in sampling methods. Could you provide more details on this?

2. Selection of Behavior Policy: How did you determine the behavior policy during training (line 220)? From lines 827 to 831, it appears that different tasks require different ratios. What standard guided these choices? The explanation on lines 835 to 840 lacks proof of the algorithm’s robustness across different ratios, which raises concerns about whether these choices were made deliberately and may impact the generality of the results. The same issue applies to the choice of C in Table 5 and α in Tables 6 and 7.

3. Choice of Backward Policy: How did you select the backward policy during training? It appears that a uniform random policy is used in the deceptive grid world and biochemical design tasks. In my understanding, the backward policy in your algorithm plays a role similar to the proposed transition kernel in MCMC, which is crucial for convergence. Could you elaborate on the role of the backward policy in your algorithm and its impact on the convergence rate of Algorithm 1 across different tasks?

4. Algorithm Performance on High-Dimensional Tasks: How does the algorithm perform in high-dimensional tasks? Sampling from a specified unnormalized density is particularly challenging in high dimensions. Testing the algorithm on high-dimensional tasks could strengthen the evaluation of its effectiveness.

5. Performance in the Manywell Task: In the Manywell task, the performances of PER+LS and the Teacher-based methods appear similar (table 8). What is the underlying intuition for this?

I would be willing to increase the score if the above questions could be well clarified.

---

> ### Author Response · Authors · 2024-11-20
> **Response (1/3)**
>
> Thank you for your detailed feedback and effort in reviewing our paper. We have attempted to answer your questions, provide new experimental results to support our claims, and clarify some possible misunderstandings below.
>
> ### W1 (lack of theoretical analysis)
>
> Please see Q1 below.
>
> ### W2 (lack of explanation of the architecture design and hyperparameters)
>
> As you mentioned, the explanation of the architecture design is crucial for greater clarity and justifiction. We mention details on implementation for each experiment in Appendix C. Here we explain more details on model architecture for each experiment.
>
> **Deceptive Grid World:** We use a two-layer MLP with 256 hidden units for the parameterized policy $P_F(\cdot;\theta)$ along with a learnable parameter for $\log Z_\theta$. The backward policy $P_B$ is fixed to a uniform policy.
>
> **Diffusion Sampling:** We employ the same architecture as [1, 2]. We encode the diffusion timestep $t$ with 128-dimensional harmonic (Fourier) features use 2 linear layers with $N=64$ hidden units to extract the signal ($x$) feature. We use a two-layer MLP with $N$ hidden units to extract feature for $x_t$ and concatenate it with the signal feature. Finally, we apply a three-layer MLP with $N$ hidden units to this concatenated representation to get $u(x_t, t;\theta)$. We initialize $\log Z_{\theta}$ to 0 for all methods. For the Manywell task, we increase $N$ from 64 to 256 to accommodate the high-dimensional tasks, and apply this adjustment to all baselines.
>
> **Biological and Chemical Discovery:** We use a similar setting to that proposed by [3]. To parameterize the forward policy, we adopt a relative edge flow policy parametrization mapping (SSR) from [3]. For QM9 and sEH tasks, we employ a two-layer architecture with 1024 hidden units, while for the other tasks, we choose to use a two-layer architecture with 128 hidden units. We initialize $\log Z_{\theta}$ to 5.0 for all methods. For the backward policy, we use a fixed uniform policy.
>
> [1] Sendera, Marcin, et al. "Improved off-policy training of diffusion samplers." The Thirty-eighth Annual Conference on Neural Information Processing Systems.
>
> [2] Zhang, Qinsheng, and Yongxin Chen. "Path Integral Sampler: A Stochastic Control Approach For Sampling." International Conference on Learning Representations.
>
> [3] Shen, Max W., et al. "Towards understanding and improving GFlowNet training." International Conference on Machine Learning. PMLR, 2023.
>
> Concerns about hyperparameters are addressed in the questions below.
>
> ### Q1 (convergence rate)
>
> Our paper makes an methodological and empirical contribution. We believe that our numerical results are sufficient to validate our proposed idea, which we expect to be influential for machine learning researchers interested in the practical aspect of GFlowNets, e.g., their application to drug discovery [1], biological sequence design [2], and language models [3,4]. To show its wide applicability, we tested the proposed algorithm on 10 tasks from different domains and found improvements over exploration methods from past work in a wide range of setting.
>
> Nevertheless, we appreciate your emphasis on the importance of convergence rates in evaluating sampling algorithms. Providing a theoretical convergence rate for Algorithm 1 is indeed valuable; however, it presents significant challenges due to the complexity inherent in deep learning-based methods. Convergence rates of deep learning models **even under a fixed sampling distribution** remains an difficult problem in the field, not to mention RL agents with a dynamic behavior policy updated in a bi-level optimization procedure.
>
> [1] Shen, Tony, *et al.* "TacoGFN: Target-conditioned GFlowNet for Structure-based Drug Design". Transactions on Machine Learning Research, 2024.
> [2] Jain, Moksh, *et al.* "Biological sequence design with gflownets." International Conference on Machine Learning (ICML), 2022.
> [3] Hu, Edward J., *et al.* "Amortizing intractable inference in large language models." International Conference on Learning Representations (ICLR), 2024.
> [4] Lee, Seanie, et al. "Learning diverse attacks on large language models for robust red-teaming and safety tuning." arXiv preprint arXiv:2405.18540, 2024.

---

> > ### Author Response · Authors · 2024-11-20
> > **Response (2/3)**
> >
> > ### Q2 (hyperparameters)
> >
> > Thank you for your insightful questions regarding the selection of the behavior policy during training and the choice of hyperparameters in our experiments.
> >
> > **Selection of Behavior Policy Ratios:**
> >
> > Firstly, we acknowledge that the behavior policy ratios are crucial hyperparameters that influence the exploration-exploitation trade-off during training. These ratios determine the proportion of samples obtained from the teacher policy, the student policy, and the replay buffer.
> >
> > Our approach involves carefully selecting these ratios based on the specific characteristics of each **domain** or **task**. While each task represents a significant and distinct domain (e.g., diffusion sampler, biochemical design), we ensure consistency by using identical hyperparameters across all subtasks within each domain. For instance, in the biochemical design tasks—including DNA, RNA, atomic molecule, and fragment molecule generation—we use the same behavior policy ratios and hyperparameters. This consistency demonstrates the stability and robustness of our algorithm within each domain.
> >
> > The choice of behavior policy ratios is guided by the nature of the task:
> >
> > - **Teacher Ratio:** Increased when exploration and mode discovery are critical, as the teacher helps the model explore diverse and high-reward regions of the state space.
> > - **On-Policy (Student) Ratio:** Increased when the task has a steep reward landscape but is less multimodal, allowing the model to exploit known high-reward areas more effectively.
> > - **Replay Buffer Ratio:** Increased when sample efficiency is important, enabling the model to learn from past experiences and improve sample utilization.
> >
> > Note that, in the  hypergrid task, we experimented with different behavior policy ratio of (teacher, student, buffer), specifically (1,1,1) and (1,1,0). Both configurations significantly outperformed the baselines in terms of number of modes discovered. The $L1$ distance tends to worsen when the buffer is utilized, whether or not Teacher is used. This is likely due to the regularization effect introduced by the highly diverse experiences provided by the buffer.
> >
> > **Minor note:** We mistakenly reported the $L_1$ value for $\alpha=0.5$ and $(d=4, H=32)$ in Table 6. We fixed it in the revised version.
> >
> > **Choice of $\alpha$:**
> >
> > Similar to the behavior policy ratios, $\alpha$ also balances between exploration and exploitation, in a different way. A high value of $\alpha$ makes the Teacher focus on high-reward areas, and thus promotes exploitation. This can be especially effective in environments with a vast search space where most regions have low rewards, as it allows the teacher to ignore the low-reward areas.
> >
> > For this reason, we set $\alpha = 0$ for the hypergrid task, where exploration is more critical by design, and $\alpha = 0.5$ for other tasks. It’s important to note that these choices were based on simple intuition rather than an extensive hyperparameter search.
> >
> > We studied the effect of $\alpha$ in Appendix D.4, providing empirical evidence for the necessity of this hyperparameter. Notably, $\alpha = 0.5$ also performs reasonably well in the hypergrid task, suggesting that it could be a good starting point for new environments.
> >
> > **Choice of $C$:**
> >
> > Regarding the hyperparameter $C$, our experiments in the hypergrid task show that varying $C$ does not significantly impact the performance of our algorithm (Appendix D.3, Table 5). We consistently outperform other baselines across different values of $C$, demonstrating the robustness of our method to this parameter.
> >
> > We appreciate your feedback, as it has allowed us to clarify these important aspects of our work.

---

> > > ### Author Response · Authors · 2024-11-20
> > > **Response (3/3)**
> > >
> > > ### Q3 (choice of backward policy)
> > >
> > > We believe there is a possible misunderstanding here. The backward policy in GFlowNets and the transition kernel in MCMC do not have a similar role. THe backward policy models 'destructive' transitions from terminal states through intermediate (incomplete) states, representing a posterior distribution over the 'constructive' sequences modeled by the forward policy starting at the initial state, passing through incomplete states, and reaching a terminal state. On the other hand, an MCMC kernel on a space models transitions between two successive complete states in a Markov chain. Put simply, MCMC is used for local exploration, while the backward policy specifies a distribution over ways to construct a given object.
> > >
> > > In GFlowNets, it is quite common to fix the backward policy to a uniform distribution. We also provide new experimental results on a **learned** backward policy:
> > >
> > > | $d=2, H=256$ | #modes | $L1$ dist. |
> > > | -------- | -------- | -------- |
> > > | TB (on-policy)     | 1289.0 $\pm$ 75.2 | 1.41 $\pm$ 0.10 |
> > > | + GAFN             | 1361.0 $\pm$ 27.3 | 1.31 $\pm$ 0.05 |
> > > | + PER              | 2139.3 $\pm$ 224.1 | 1.44 $\pm$ 0.06 |
> > > | + Teacher (Ours)   | **2149.3** $\pm$ 89.4 | **1.25** $\pm$ 0.04 |
> > >
> > > | $d=4, H=32$ | #modes | $L1$ dist. |
> > > | -------- | -------- | -------- |
> > > | TB (on-policy)     | 9.7 $\pm$ 1.2 | **1.632** $\pm$ 0.000 |
> > > | + GAFN             | 18.7 $\pm$ 2.9 | 1.650 $\pm$ 0.003 |
> > > | + PER              | 20.7 $\pm$ 7.4 | 1.636 $\pm$ 0.000 |
> > > | + Teacher (Ours)   | **226.7** $\pm$ 13.2 | 1.633 $\pm$ 0.000 |
> > >
> > > These results show that our claims continue to hold in the setting of a learned backward policy.
> > >
> > > ### Q4 (high-dimensional tasks)
> > >
> > > Thanks for pointing this out. Although the benchmarks we considered are quite standard in the sampling literature, cf. [Sendera et al., 2024] -- note that sampling an unnormalized density with no prior information is much more difficult than generative modeling given data -- we also test our algorithm in terms of scaling in two additional experiments: (1) scaling the combinatorial space of the hypergrid and (2) application to a real-world LLM red-teaming benchmark.
> > >
> > > For (1),(2) we get better performances than baseline GFlowNets and replay buffer-based off-policy training methods. The results for (1) are shown below; for plot-based results on (2), please see Appendix F of the revised manuscript.
> > >
> > > **(1) The results on larger hypergrid**
> > > We test in a grid setting $(d=4, H=128)$, where the total number of terminal states $\vert \mathcal{X} \vert = 268,338,173$. Note that we report only the number of modes discovered (#modes) since we can't calculate $L1$ distance in larger problems due to the computational burden for obtaining the target distribution analytically.
> > >
> > > | $d=4, H=128$ | #modes |
> > > | -------- | -------- |
> > > | TB (on-policy)     | 228.7 $\pm$ 38.1 |
> > > | + GAFN             | 180.0 $\pm$ 51.9 |
> > > | + PER              | 164.3 $\pm$ 23.7 |
> > > | + Teacher (Ours)   | **728.0** $\pm$ 192.0 |
> > >
> > > From this result and Table 1 of the manuscript, we can see that the relative performance of Teacher gets better as the dimension increases, showing its scalablility.
> > >
> > >
> > > ### Q5 (performance on ManyWell)
> > >
> > > We suspect there may be a small misunderstanding in the interpretation of the results. The ELBO and EUBO should be below and above the true $\log Z$ value, respectively, which in this task is ~164.696. The results should be understood in terms of the *distance from their true value*: in fact, the proposed algorithm's performance is **very close to the optimum** and **significantly better than PER+LS**.
> > >
> > > We also note that in the Manywell task, the local search (LS) method combined with Prioritized Experience Replay (PER) employs the Metropolis-adjusted Langevin algorithm (MALA). MALA is a powerful sampling technique because it utilizes gradient information of the energy function to guide the search process effectively.
> > >
> > > Our Teacher-based method, as described in Task 2, is designed for general black-box reward and energy functions and does not rely on gradient information. The fact that our Teacher-based method slightly outperforms PER+LS -- even though both methods nearly achieve optimal sampling -- is particularly noteworthy. This is promising because our method achieves comparable or better results without the additional assumptions and computational overhead required by MALA, such as access to the energy function's gradients. **This outcome demonstrates the effectiveness of our approach in efficiently sampling from complex distributions even when gradient information is unavailable.**
> > >
> > > **Thank you again for your comments. We hope our answers have helped to resolve many of your concerns about our work, and we are happy to answer any further questions you may have during the discussion period.**

---

> > > > ### Author Response · Authors · 2024-11-22
> > > > **Follow-up**
> > > >
> > > > Dear Reviewer CGLL,
> > > >
> > > > We just want to let you know that we've updated Appendix F with several larger-scale deceptive grid world results and the experiment on LLM red-teaming, in response to your questions about scaling performance. We hope these help to address your concerns and look forward to your feedback.
> > > >
> > > > The authors

---

> ### Author Response · Authors · 2024-11-25
>
> Dear Reviewer CGLL,
>
> As we reach the end of the discussion period, we'd like to ask if we can provide any more information that could affect your assessment of the paper. In the discussion period, we believe that we clarified the concerns and questions raised. Specifically, we included additional experiments to address concerns about scalability and flexibility under different backward policy settings (the learned $P_B$). This will be a good improvement to our manuscript; thanks again for your feedback.
>
> The authors

---

> > ### Comment · Reviewer_CGLL · 2024-11-26
> > **Increase the score**
> >
> > Thanks to the author for their response and explanation. Based on the changes, I decided to increase my score to 5.

---

### Official Review · Reviewer_NTRq · 2024-10-31

**Soundness:** 4
**Presentation:** 4
**Contribution:** 4
**Rating:** 8
**Confidence:** 5

**Summary:**

This work focuses on the efficient exploration of RL training during amortized inference. The primary contribution lies in developing an adaptive training distribution to guide the amortized sampler in prioritizing difficult ones. The proposed method is examined in a collection of benchmarks, including both synthetic and real-world scenarios. The exploration efficiency and other benefits are reflected in both mode coverage and sample efficiency.

**Strengths:**

I can easily follow this work, and this work tries to amortize prediction by simply conditionally inputting some variables.

Overall, (1) I find this work easy to follow with clear motivations. Decision-making for amortized inference, particularly the development of GFlowNets, is impactful, and this work focuses on an important issue, namely efficient exploration under RL frameworks in the field. (2) The developed strategy is novel and practical in implementation. (3) The experiments are inspiring and well-supported claims.

**Weaknesses:**

While a lot of merits in this work, I find some parts are necessary to modify or revise.

---

(1) It seems to lack the necessity of amortized inference. In line28-30, it states the mechanism of amortized inference and related bottleneck. It is necessary to include the role of amortized inference compared with traditional methods such as MCMC, e.g., citing [1] and adding something like

"The amortized inference adopts a shared inference module for all data points instead of performing inference one by one. In this way, we can reuse the computational module for other data point's inference."

(2) There exists literature work [2] that raises similar learning modules in terms of training adaptative distributions for few-shot experimental design; it is necessary to discuss them in detail in Section 4 related work.


(3) Other suggestions or questions: (i) Figure 1 is clear enough, but I am not sure whether there should exist links between the Buffer and the Teacher or the Student to reveal the data flow. (ii) in Line-74, it says "we believe that trajectories with high loss are particularly informative for mode coverage", is this a hypothesis? Are there any explanations from either experiments or other intuitions? (iii) In Line 60, it says "the Student's target distribution does not depend on the Teacher", hence I am wondering whether the optimization pipeline is an adversarial game. (iv) In line 237, I want to know how to balance the student, the teacher or a buffer.


**Reference:**

[1] Margossian C C, Blei D M. Amortized Variational Inference: When and Why?[J]. arXiv preprint arXiv:2307.11018, 2023.

[2] Wang, Cheems, et al. "Robust Fast Adaptation from Adversarially Explicit Task Distribution Generation." arXiv preprint arXiv:2407.19523 (2024).

---

I will be happy to update my score if these concerns are well addressed during the rebuttal discussion.

---

Post Rebuttal

The author has well addressed my concerns, and I updated my score to accept.

**Questions:**

See Weakness part

---

> ### Author Response · Authors · 2024-11-20
> **Response**
>
> Thank you for your review and the positive assessment of our paper. We answer your concerns below.
>
> ### Why amortized inference?
>
> Thank you for your suggestion. We agree that the ability to reuse a shared computational module for inference across multiple data points, as opposed to performing inference independently for each data point, is a major motivation for amortized inference compared to MCMC. Following your recommendation, we have cited the suggested reference and revised the first paragraph of introduction to better highlight this key advantage of amortized inference.
>
>
> ### Related work on experimental design
>
> Thank you for pointing out this relevant literature. We agree that it is related to our work, as our Teacher plays the same role as the entropy regularized adversary that generates tasks. We have included discussion of the suggested reference in the related work (Sec. 4).
>
> ### Figure 1
>
> The illustration is intended to show that the behavior policy (Teacher, Student, or Buffer) contributes to the data flow for both Teacher and Student training. This implies that the Buffer sometimes provides data to the Teacher and Student during training.
>
> In the revised version, we have added an arrow from the Student to the Buffer to Figure 1, as trajectories are collected from both models.
>
> ### Why are high-loss trajectories informative?
>
> This is a hypothesis, motivated by the following arguments:
> - In all learning systems, samples with high error, where the current iteration of the model struggles, tend to be more informative. This is a principle used in active learning, hard example selection, curriculum learning, prioritized experienced replay in RL, etc.
> - For amortized inference systems in particular, discovery of poorly modeled modes (especially those whose density is underestimated) is critical. This is because errors in *already observed* modes can self-correct, as they are revisited during training, but missing modes are unlikely to be visited by on-policy (or near-on-policy) sampling, making them harder to recover.
>
> Thus, to promote discovery of modes, the Teacher should guide the Student toward samples where the sampling density divergences from the target, especially favouring the samples whose density is underestimated. This motivates the proposed reward for the Teacher (equation 5).
>
>
> ### Is this an adversarial game?
>
> Good question. Assuming the function classes of the Student and Teacher policies can express the unique point where both achieve zero loss, the joint learning problem between Teacher and Student is not adversarial in the sense of having a saddle point at the optimum. The losses for both models are strictly positive and are zero precisely at the optimum (cf. Proposition 1 in Appendix A), so any deviation from the optimum will not decrease the losses of teacher, student, or both.
>
> Furthermore, the Student's reward does not depend on the parameters of the Teacher, even though its *training policy* is, which implies that **the optimality of a Student policy is independent of the Teacher's parameters**.
>
> However, if the optimal policies are not representable by the policy networks of the Student and Teacher, a saddle point at the optimum is possible.
>
> ### Details of behavior policies
>
> The details are provided in Appendix B (lines 827-831). The balancing rule is straightforward: the Teacher focuses on multi-modal exploration, the Student emphasizes exploitation, and the replay buffer is used for sample efficiency. Depending on the characteristics of the target task, users can adjust the balance according to their specific needs, similar to how exploration-exploitation trade-offs are tuned in other exploration methods in RL.
>
> **We appreciate your valuable input; please let us know if we can provide any further clarifications.**

---

> ### Comment · Reviewer_NTRq · 2024-11-21
> **Good paper and updated the score**
>
> Thanks for the author's detailed response and clarifications.
> After revision, the updated manuscript is complete enough.
> Taking other reviewers' comments and my assessments, I think this work is well-motivated and proposes a novel approach to Bayesian deep learning with sufficient evaluation.
> Hence, I have updated my score and think this work deserves acceptance.

---

### Official Review · Reviewer_y6wN · 2024-11-05

**Soundness:** 3
**Presentation:** 3
**Contribution:** 3
**Rating:** 8
**Confidence:** 3

**Summary:**

The paper presents a novel method to improve amortized inference for complex distributions using an adaptive "Teacher-Student" training framework. The Student is an amortized sampler parameterized as a generative flow network (GFlowNets) and trained using RL. The primary contribution of the work is introducing the Teacher as an auxiliary  model that acts as the 'exploration policy' for the student. It is trained to guide the Student training by focusing on high-loss regions, thereby promoting the discovery of unexplored modes of the target distribution. The proposed method is evaluated on synthetic environments, diffusion-based sampling tasks, and biochemical discovery tasks, demonstrating improved mode coverage and sample efficiency.

**Strengths:**

**Strengths** :
- Addresses an important problem: Mode coverage/exploration is an important problem in the training of GFlowNets. The paper proposes a novel and interesting solution to the problem.

- Very Well written :  I really enjoyed reading the paper. The paper did an excellent job of introducing and walking through the relevant literature and the methods and putting itself in context. Although I was not myself very familiar with the specific work line of work around GFlowNets, I was easily able to follow along all the details.

- Lots of interesting details : The training formulation was interesting, especial given I wasn't very familiar with the glownets literature before this. I also particularly liked the use of a search procedure combined with the Teacher network to reduce the teacher network induced bias in the exploration process (although this was already introduced in previous work!). This approach effectively guides the student towards more diverse solutions, improving the overall learning efficiency and robustness.

- Impressive empirical results: The paper demonstrates the versatility of the Teacher-Student framework by applying it to a range of tasks, including synthetic benchmarks and biochemical discovery problems. The empirical results consistently show that the Teacher-Student setup leads to better mode coverage and training efficiency compared to existing methods. Especially the results in more complicated tasks with a large number of modes.

**Weaknesses:**

**Weaknesses/Questions**
-  The introduction of an adaptive Teacher adds additional complexity to the training process, requiring the joint optimization of both Teacher and Student networks. At least in the RL literature, these types of exploration methods were tried and given up on as they required extensive tuning and didn't scale well enough. I'm curious how the authors think that compares with the use cases here and if the authors genuinely believe the results shown in the paper will hold the test of time?

- This is maybe a dumb question. But I do wonder how these methods compare with using standard sampling based strategies for mode discovery e.g  thompson sampling etc. My understanding is those become intractable as the problem size increases. The approach suggested seems pretty complex and I do wonder if the those standard mode discovery methods could've helped make things simpler.

- Details on the biochemical discovery experiments were a little unclear. Eg. How do you define the reward function etc was not very clear to me.

**Questions:**

same as weaknesses

---

> ### Author Response · Authors · 2024-11-20
> **Response (1/2)**
>
> Thank you for such a positive assessment of our paper's relevance, exposition, and empirical evaluation. We've attempted to answer your questions below.
>
> ### Complexity of training and simpler exploration strategies
>
> The question of whether the benefits brought by the proposed algorithm are worth the added complexity of training a teacher model, relative to simpler exploration methods, is very important, and we appreciate the opportunity to comment upon this point.
>
> Certainly, the additional tunable parameters are unavoidable when we introduce a new component (like the Teacher network) to an algorithm. However, these parameters are often beneficial in that they provide controllability of a desired behavior, in our case, exploration. This can be compared to the way that GANs improve the training of a generator by introducing a secondary network, the discriminator, into the optimization.
>
> Although the joint optimization of two networks increases training complexity, there are reasons to believe the benefits outweigh the shortcomings, even at large scales. It is clear that large-scale tasks where exploration is difficult require a well-designed training policy for an agent. The more complex the task, the harder it is for a replay buffer to capture all modes of the target distribution. The Teacher network can be viewed as an amortized replay buffer that provides generalization abilities that a buffer cannot -- it can generate arbitrarily many new training samples for the Student on the fly. This approach is fundamentally more scalable compared to non-learned sample selection methods.
>
> As for comparison to simpler exploration methods:
> - We compared our method to simple approaches such as epsilon-greedy and replay buffer methods, but they did not achieve satisfactory performance, particularly in large-scale tasks.
> - Methods like Thompson sampling, which rely on maintaining a Bayesian posterior over parameters, become extremely complex at scale, while RND requires training a second model, typically of a similar complexity to the sampler (just as our Teacher). Note that we also compare with both of these methods in the form they were proposed in prior work, but again found them to underperform.
>
> Finally, we want to emphasize that methodological simplicity is not equivalent to the complexity or scalability of an algorithm. While you mention that our method requires joint optimization and may seem complex, this does not necessarily mean it introduces significant complexity at scale. Our approach simply involves training one additional network at any scale, leading to a constant multiplicative increase in number of parameters.

---

> > ### Author Response · Authors · 2024-11-20
> > **Response (2/2)**
> >
> > ### Missing details in biochemical tasks
> >
> > We apologize for the unclear explanation of the biochemical discovery experiments. We explain how we define the reward function below and have added these details to the manuscript (Sec. 5.3).
> >
> > **QM9:** Our goal is to generate a molecular graph. The reward function is a HOMO-LUMO gap on the target transcription factor, which is obtained via a pre-trained MXMNet proxy from [1]. We use a reward exponent of 5. We define modes as the top 0.5% quantile of $R(x)$.
> >
> > **sEH:** Our goal is to generate binders of the sEH protein. The reward function is a binding affinity to soluble epoxide hydrolase (sEH), which is provided by the pre-trained proxy model from [2]. We use a reward exponent of 6. We define modes as the top 0.01% quantile of $R(x)$, with additional filtering to exclude candidates that are too similar to each other based on Tanimoto similarity, following [3].
> >
> > **TFBind8:** Our goal is to generate a DNA sequence of length 8. The reward function is a binding affinity to a human transcription factor, which is obtained via a pre-trained proxy model provided by [4]. We use a reward exponent of 3. We use a pre-defined set of modes provided by [5].
> >
> > **L14-RNA1:** Our goal is to generate a RNA sequence of length 14. The reward function is a binding affinity to a human transcription factor, which is obtained via a pre-trained proxy model provided by [6]. We use a reward exponent of 8. We define modes as the top 0.01% quantile of $R(x)$ and the diversity threshold as 1 unit of Levenstein distance, also following [3].
> >
> > [1] Zhang, Shuo, Yang Liu, and Lei Xie. "Molecular mechanics-driven graph neural network with multiplex graph for molecular structures." arXiv preprint arXiv:2011.07457, 2020.
> > [2] Bengio, Emmanuel, *et al.* "Flow network based generative models for non-iterative diverse candidate generation." Neural Information Processing Systems (NeurIPS), 2021.
> > [3] Kim, Minsu, *et al.* "Learning to scale logits for temperature-conditional GFlowNets." International Conference on Machine Learning (ICML), 2024.
> > [4] Trabucco, Brandon, *et al.* "Design-bench: Benchmarks for data-driven offline model-based optimization." In International Conference on Machine Learning (ICML), 2022.
> > [5] Shen, Max W., *et al.* "Towards understanding and improving GFlowNet training." International Conference on Machine Learning (ICML), 2023.
> > [6] Sinai, Sam, *et al.* "Adalead: A simple and robust adaptive greedy search algorithm for sequence design." arXiv preprint arXiv:2010.02141, 2020.
> >
> > **Thank you again for your feedback and interesting comments. We are happy to respond to any further questions you may have.**

---

### Official Review · Reviewer_tjcQ · 2024-11-06

**Soundness:** 2
**Presentation:** 2
**Contribution:** 2
**Rating:** 5
**Confidence:** 3

**Summary:**

This paper proposed a method to improve mode coverage and training efficiency in amortized inference methods like GFlowNets. Specifically, the authors use offline RL training to encourage the discovery of diverse, high-reward candidates, and addressed the key challenge -- exploration in off-policy RL. The main idea is to use an adaptive "Teacher" model to help the "Student" sampler by focusing on regions with high loss. The teacher model is used as as an auxiliary model, is trained to target areas where the Student model has high errors. This allows it to cover unexplored modes (which usually have high errors) and provide a more efficient training process. Empirically, the authors show that this approach works well in various tasks, such as discrete sequence design and continuous diffusion sampling, with better sample efficiency and mode coverage.

**Strengths:**

1. The idea of exploring high-error region and increase the sampling probability of data in these region for training the student model intuitively makes sense to me, which is essentially resembles to hard-negative mining in the classic machine learning literature.

2. The experiments were well-executed and supports the main claim in the paper.

3. The math on GFlowNets and their connection to amortized inference is helpful, especially helps contextualize the significance of the contributions.

**Weaknesses:**

1. The idea is not new; it closely resembles hard negative mining (i.e., sampling negative examples where the model shows high error), which limits the novelty of the proposed approach.

2. While the idea of sampling more in high-error regions seems intuitively reasonable, its effectiveness may depend on whether the student model has sufficient capacity to fit the distribution. Also, I would like to see more comparisons and discussion with the active learning literature, such as uncertainty sampling, etc.

**Questions:**

Can the author explain why you chose GFlowNets for the experiments? Are they more effective than diffusion models for chemical or drug discovery? In my understanding, diffusion models can fairly easily to fit multimodal distributions.

---

> ### Author Response · Authors · 2024-11-20
> **Response (1/2)**
>
> Thank you for your review. We appreciate that you described the proposed algorithm as intuitive and found the exposition and experiments clear and well-executed. Below we answer the questions and concerns you raised.
>
> ### Novelty and resemblance to hard negative mining
>
> Our main idea is to **amortize** the sampling of high-loss examples through the use of a teacher network, rather than selecting hard examples from a dataset. This clearly distinguishes our work from hard example mining [[Robinson et al.](https://openreview.net/forum?id=CR1XOQ0UTh-), 2021].
>
> In the RL setting (of which GFlowNets are one instance), hard example mining is related to prioritized experience replay (PER), in which a buffer functions as the dataset from which hard examples are chosen. We indeed consider PER as a baseline in our study.
>
> ### Model capacity
>
> We agree that the effectiveness of the sampler resulting from our proposed algorithm depends on the student model's capacity to fit the distribution. This is a consideration common to all training methods for amortized samplers and involves such questions as the neural network architecture and the update rule (optimizer) used for the policy network for each given sample.
>
> However, **the problem we address is orthogonal**: for a student model of given architecture, how do we best select the *training distribution* (i.e., behavior policy) that provides it the samples to learn from? Our solution, which introduces an auxiliary teacher model, leads to improved exploration relative to other techniques that use the same student model architecture. In fact, our technique *decouples* the modeling capacity of the student from the behaviour policy, unlike other methods (noisy on-policy, Thompson sampling, etc.) that use modifications to the student's policy to induce exploration.
>
> ### Comparison with active learning
>
> Thank you for pointing out the connection with active learning. We agree that there are strong connections between active learning and our method, as both aim to leverage information about regions where the current model performs poorly. However, there are also clear distinctions between the problems considered in the two areas.
>
> - **Active learning** is primarily a framework for supervised learning, where the goal is to select the most informative data points for labeling. These methods typically rely on a form of *predictive uncertainty* to select inputs $x$ that optimally inform the learning of a mapping to labels $y$, using information derived from the classifier's probabilistic model $p(y\mid x)$ but **without seeing the true label $y$**. Some forms of uncertainty used to guide example selection include margin sampling, maximum-entropy sampling, etc., as well as Bayesian uncertainty quantification approaches (ensembles and BNNs, *inter alia*), all of which query the samples $x$ where the predictor is, in some sense, most uncertain of the label.
> - In contrast, **our method** is based on reinforcement learning (RL), not supervised learning. Instead of training a classifier to model $p(y\mid x)$, we train a sequential decision-making agent characterized by a policy $P_F(\tau) = \prod_{t=1}^n p(s_t|s_{t-1})$, where the trajectory $\tau$ represents a sequence of states $(s_0, s_1, \ldots, s_n)$. Gradient updates are made using a loss that depends on a trajectory $\tau$, which is not necessarily sampled from the policy itself. Our solution to the trajectory selection problem does notes not rely on predictive uncertainty. Instead, it leverages information from the *loss values*, which are computed using the true terminal reward values.
>
> The two problems have fundamentally different characteristics, despite their conceptual similarities: in active learning, one seeks areas with high uncertainty (that is, possibly high *unknown* loss value), while we amortize the sampling of areas with high (*known*) loss value.
>
> In the revised manuscript (Sec. 4, Related Work), we have included a more detailed comparison with the active learning literature, including uncertainty sampling methods.

---

> > ### Author Response · Authors · 2024-11-20
> > **Response (2/2)**
> >
> > ### Comparison with diffusion models (including in biochemical tasks)?
> >
> > Firstly, it's important to clarify that diffusion models (as the term is typically understood) are generative models, usually trained to maximize a variational bound on log-likelihood of a dataset. GFlowNets, on the other hand, are **training methods** that fit the parameters of any sequential generative process to sample a given density function (reward), absent a dataset.
> >
> > Because diffusion models assume a sequential generative process, GFlowNet algorithms can indeed be used to train diffusion models *without data samples, but given a target energy that we wish to sample*. In fact, this is done in our second task, the "diffusion sampler", and most fully explored in the reference [Sendera et al., 2024]. In our work, we showed that the proposed Teacher-Student method can be used to improve the training of diffusion samplers.
> >
> > Regarding chemical or drug discovery, our benchmarks are built directly on the previous work by [Shen et al., 2023], where a bidirectional sequence generative model was trained using GFlowNet algorithms. For a fair comparison, we used the same generative model architecture but explored different GFlowNet training methods. As far as we are aware, discrete diffusion models have not previously been applied to this particular task, and comparing the effectiveness of discrete diffusion models versus other generative models is not related to the focus of our work. Our goal is to improve exploration in GFlowNets as a training method, regardless of the underlying generative model.
> >
> > **Thank you again for your comments. We hope we have addressed them satisfactorily above, but do not hesitate to let us know if you have further questions.**

---

> > > ### Author Response · Authors · 2024-11-25
> > >
> > > Dear reviewer tjcQ,
> > >
> > > As we reach the end of the discussion period, we’d like to ask we can provide any more information that could affect your assessment of the paper. We believe that our answer above has addressed your original concerns and clarified a few points that may have been missed. Thanks again for your attention to our work.
> > >
> > > Best,
> > > Authors

---

> > > > ### Comment · Reviewer_tjcQ · 2024-11-28
> > > >
> > > > Thanks for the authors responses. The authors response addressed many of my concerns. However, I am skeptical about the practicality of the GFlowNets, especially given the wide adoption of diffusion models in many domains, such as image/video/audio generations, protein design, drug discovery etc. The training of diffusion models is actually quite simple and scalable. Maybe I am missing something, could the author explain what's hindering the adoption of Gflownets in various applications? I am borderline to this paper.

---

> ### Author Response · Authors · 2024-11-29
> **Diffusion models and GFN are orthogonal methods; combining them has proven to be useful.**
>
> We believe there is a slight misunderstanding. We agree that diffusion models are very useful and that we must continue researching how to improve such models.
>
> **Diffusion models and GFNs are orthogonal**: A GFN is a training method for diffusion models without data, but with energy/reward.
>
> Training diffusion models with denoising score matching (DSM) or maximum likelihood estimation (MLE)—typical methods for diffusion models—is scalable and practical when massive datasets are available. However, when we aim to enhance diffusion models with a reward model or perform intractable inference using energy functions, we need reinforcement learning (RL)-like training methods because there is no such massive dataset to imitate (making DSM or MLE on a dataset impossible). Among these methods, GFN are a promising candidate.
>
> For example, Venkatraman et al. [1] demonstrated that fine-tuning diffusion models with a reward model can be effectively applied to various tasks, including **inverse image problems**, **language model infilling using discrete diffusion**, **text-to-image model fine-tuning**, and **offline RL** through GFN training over diffusion models.
>
> Moreover, Seong et al. [2] shows that using the same GFN objective with Venkatraman et al. [1], we can model **molecular dynamics (MD)** by sampling rare transition paths. In such scientific discovery applications, there are desired models called **Boltzmann Generators** that aim to sample proportionally to the Boltzmann energy distribution $e^{-E(x)/T}$ (in MD and N-body particle simulations). For Boltzmann Generator training, we believe this combination is particularly useful: (1) we have to use diffusion models, and (2) train them using a GFN-like objective. There is active research following (1) and (2) to meet these demands [3, 4].
>
> ---
>
> [1] Venkatraman et al. "Amortizing Intractable Inference in Diffusion Models for Vision, Language, and Controls", NeurIPS 2024.
>
> [2] Seong et al. "Collective Variable Free Transition Path Sampling with Generative Flow Network", ICML Workshop on Structured Probabilistic Inference and Generative Modeling 2024.
>
> [3] Sendera et al. "Improved Off-policy Training of Diffusion Samplers", NeurIPS 2024.
>
> [4] Akhound-Sadegh et al., "Iterated Denoising Energy Matching for Sampling from Boltzmann Densities", ICML 2024.

---

> > ### Author Response · Authors · 2024-12-02
> >
> > Dear Reviewer tjcQ,
> >
> > Just a gentle reminder that our discussion period is ending soon. If you feel that your concerns have been addressed in our previous responses, please consider revising the score. If there are any remaining issues, please let us know as soon as possible.
> >
> > Best regards,
> > The Authors

---

### Author Response · Authors · 2024-11-22
**New experiments (scaling, LLMs) and changes to the paper**

Dear reviewers,

We'd like to thank you again for your effort in reviewing our paper and the thoughtful comments you've made, particularly Reviewer NTRq, who has already answered our initial comment.

We have responded individually to your feedback, but would like to use this message to summarize the main changes to the paper (a second revision has just been uploaded) and to share some new experiment results that we believe strengthen our claims.

### Summary of main changes

- **Introduction**: In the first paragraph, we added a brief explanation of why amortized inference is essential, as suggested by NTRq.
- **Figure 1**: We included an arrow from the Student to the Buffer to more accurately represent the algorithm, again based on NTRq’s feedback.
- **Related Work**: We added additional references and discussions on active learning (thanks to tjcQ) and experimental design (thanks to NTRq).
- **Section 5.3**: We provided more detailed descriptions of the experimental settings for biochemical discovery, as requested by y6wN.

### New results

- **Appendix F**: In response to the feedback from CGLL and y6wN, we conducted additional experiments to validate the scalability of our algorithm. These experiments include:
  - a large-scale deceptive grid world (Appendix F.1);
  - the task of sampling attack prompts on pretrained LLMs (Appendix F.2).

These new experiments illustrate that the proposed algorithm remains effective in combinatorially complex environments (hypergrid) and with large models on real-world tasks (LLM).

**Thanks again, and please let us know if you have any questions.**

The authors

---

### Meta-Review · Area_Chair_MSAP · 2024-12-23

**Metareview:**

In this paper, the authors improve the efficiency of training off-policy RL by developing a method to perform amortized inference in such a manner as to prioritize high-loss regions of the loss.  They do this by proposing a "teacher" that creates an adaptive training distribution to prioritize high-loss regions of a "student" model.  The method is demonstrated with GFlowNets and they show that this improves mode coverage and sample efficiency.

The reviews were high variance but leaning towards accept, with two accepts and two marginal leaning reject.  Of the two more negative reviewers, one had concerns about novelty (e.g. compared to hard-negative mining and uncertainty sampling) and the other mostly asked for theoretical justification.  One of these reviewers raised their score from 3 to 5 after reading the author rebuttal.  The reviewers in general found the paper sound, well-written and well motivated.

In subsequent discussion, one of the reviewers voiced that they wished to champion the paper for acceptance noting that the method is novel, the paper very well written and that theoretical justification seems unnecessary.  When prompted, none of the reviewers voiced any concerns or disagreed.

Given that two reviewers are willing to champion accept and that the average is over the bar, the recommendation is to accept the paper.

**Additional Comments On Reviewer Discussion:**

The authors responded to all the reviewers' comments.  One of the reviewers, the one asking for theoretical justification, raised their score from a 3 to a 5.  The other 5 read the rebuttal but decided to leave their score unchanged.

In the reviewer / AC discussion period I prompted the reviewers about whether the concerns about novelty and theoretical justification were sufficient to strongly argue against acceptance (or if anyone would champion the paper).  Reviewer NTRq strongly argued for acceptance, citing the authors' responses as compelling.  None of the other reviewers chimed in.  In my view, unless the other reviewers have issues with technical correctness, attribution, or other compelling reasons, then if two reviewers argue strongly for acceptance the paper should be interesting / exciting to at least a subset of the community.

---

### Decision · Program_Chairs · 2025-01-22

Accept (Poster)